# T²PO: Uncertainty-Guided Exploration Control for Stable Multi-Turn Agentic Reinforcement Learning

**Haixin Wang** [1]   **Hejie Cui** [2] [*]   **Chenwei Zhang** [2]   **Xin Liu** [2]   **Shuowei Jin** [2]   **Shijie Geng** [2]   **Xinyang Zhang** [2]
**Nasser Zalmout** [2]   **Zhenyu Shi** [2]   **Yizhou Sun** [1]

## Abstract

Recent progress in multi-turn reinforcement learning (RL) has significantly improved reasoning LLMs' performances on complex interactive tasks. Despite advances in stabilization techniques such as fine-grained credit assignment and trajectory filtering, instability remains pervasive and often leads to training collapse. We argue that this instability stems from *inefficient exploration* in multi-turn settings, where policies continue to generate low-information actions that neither reduce uncertainty nor advance task progress. To address this issue, we propose Token- and Turn-level Policy Optimization (**T²PO**), an uncertainty-aware framework that explicitly controls exploration at fine-grained levels. At the token level, T²PO monitors uncertainty dynamics and triggers a thinking intervention once the marginal uncertainty change falls below a threshold. At the turn level, T²PO identifies interactions with negligible exploration progress and dynamically resamples such turns to avoid wasted rollouts. We evaluate T²PO in diverse environments, including WebShop, ALF-World, and Search QA, demonstrating substantial gains in training stability and performance improvements with better exploration efficiency. Code is available at: `https://github.com/WillDreamer/T2PO`.

## 1. Introduction

Recent advances in self-evolving agents are deeply rooted in multi-turn reinforcement learning (RL) (Liu et al., 2024; Team, 2025; Team et al., 2025; Wang et al., 2026), which provides the foundational mechanism for training agents to reason, act, and self-evolve through iterative interactions with the environments. Despite this progress, the community still lacks a stable and scalable training paradigm. Current multi-turn RL pipelines face intertwined challenges in both effectiveness and efficiency. On the one hand, long-horizon interactions combined with sparse reward signals make credit assignment inherently difficult (Zhou et al., 2024; Wang et al., 2025b). On the other hand, rollout collection is computationally expensive, driving the adoption of acceleration techniques such as low-precision inference (Liu et al., 2025a) and asynchronous sampling (Fu et al., 2025a). Yet these efficiency-oriented solutions inevitably introduce off-policy drift and stale policy effects (Zheng et al., 2025a). Both of these issues tend to amplify training instability and frequently lead to the notorious training collapse.

To mitigate training instability, prior work has explored a variety of strategies, including fine-grained credit assignment (Feng et al., 2025), internal or process-based reward modeling (Wang et al., 2025a; Dong et al., 2025), and trajectory-level filtering of failed interactions (Yu et al., 2025; Xue et al., 2025). These approaches aim to provide denser learning signals or remove void rollouts, and have shown partial success in stabilizing optimization. However, most existing solutions operate either at a coarse trajectory level or through implicit control via reward shaping. In the inherently complex multi-turn setting, such coarse or indirect interventions make the training dynamics highly sensitive to hyperparameters and rollout distributions. As a result, they often lead to *training collapse*, the phenomenon characterized by rapidly degrading performance or complete failure of policy optimization, as illustrated in Figure 1.

OUR KEY INSIGHT. To understand its origin, we analyze representative training trajectories and identify *insufficient exploration* as the underlying cause, reflecting a systematic violation of the exploration–exploitation trade-off (Mehlhorn et al., 2015). We refer to this failure mode as **hesitation**. At the token level, LLM agents frequently exhibit overthinking, generating long sequences of tokens whose information gain rapidly saturates, while their sampling noise continues to accumulate. At the turn level, LLM agents may

Part of this work was done during an internship at Amazon. [1]University of California, Los Angeles [2]Amazon.com Inc. Correspondence to: Hejie Cui[*] <cuihejie331771@gmail.com>.

*Proceedings of the 43rd International Conference on Machine Learning*, Seoul, South Korea. PMLR 306, 2026. Copyright 2026 by the author(s).

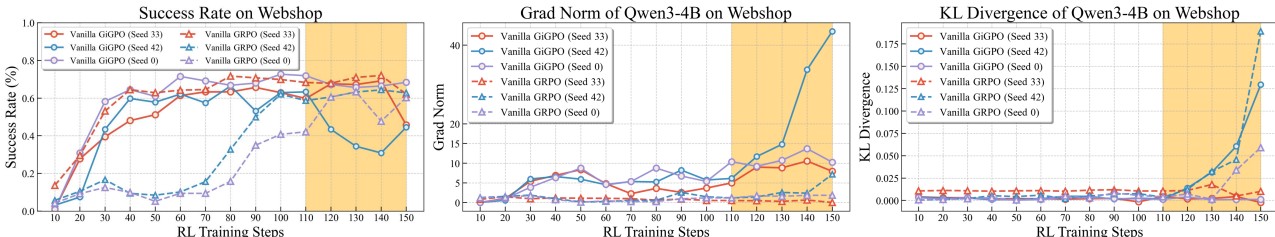

*Figure 1.* Training instability of SOTA baselines under different environment initialization random seeds. We can observe that success rate drops while internal signals like KL divergence and gradient norms explode (shown in orange background).

deviate from the successful action space at an early stage, yet continue executing numerous repetitive and unproductive turns, leaving little chance of recovery within a limited budget. Hesitation is defeat! Such behaviors introduce substantial noise into credit assignment, resulting in unstable gradients and high variance in policy updates.

Training effectiveness and efficiency need not be at odds; they can be jointly optimized once the root cause of instability is properly identified. We aim to overcome the hesitations by controlling exploration through the capture of intrinsic signals before exploration becomes inefficient. First, we construct a self-calibrated uncertainty signal by fusing entropy and confidence, which serves as a monitoring signal during rollouts. Then, we observe that continued token generation without a noticeable reduction in self-calibrated uncertainty indicates token-level hesitation, while repeated turns exhibiting similar uncertainty patterns indicate turn-level hesitation.

In this work, we propose T²PO to explicitly and finely control exploration. At the token level, T²PO monitors uncertainty dynamics and triggers thinking intervention once marginal uncertainty change falls below a threshold. At the turn level, T²PO identifies interactions with negligible exploration progress and dynamically resamples such turns to avoid wasted rollouts. By explicitly reducing inefficient exploration rather than introducing additional reward shaping, T²PO restores a balanced exploration–exploitation regime. Besides, we employ rejection-based fine-tuning (RFT) (Wei et al., 2025) for cold-start, introduce a memory context window to alleviate training pressure, enforce a strict format penalty for structural compliance, and finally adopt SOTA policy update methods for optimization.

Extensive experiments on challenging multi-turn agentic benchmarks, including WebShop, ALFWorld, and Search QA, demonstrate the superiority of T²PO. Besides, comprehensive ablations and analyzes further verify its effectiveness in improving exploration efficiency.

**Conflict of Interest Disclosure**. The authors declare no financial conflicts of interest in this paper.

## 2. Related Works

### 2.1. Agentic RL Training

Early work on LLM agents focused on modular infrastructures for interaction and evaluation. RAGEN (Wang et al., 2025b) established a unified framework for training and benchmarking agentic RL systems. Subsequent efforts sought to stabilize multi-turn training through trajectory curation and sampling. SimpleTIR (Xue et al., 2025) filters rollouts containing void turns, while rStar2-Agent (Shang et al., 2025) oversamples rollout groups and retains only high-quality trajectories, improving training stability via heuristic data selection. However, these methods rely on external filtering and do not explicitly regulate reasoning dynamics within trajectories. Meanwhile, group-based critic-free optimization has emerged as an efficient paradigm for long-horizon agent training. GiGPO (Feng et al., 2025) extends group-based advantage estimation to multi-turn settings, achieving strong performance without auxiliary value networks. Yet existing multi-turn group-based methods still lack principled mechanisms to suppress redundant reasoning within and across turns, resulting in inefficient exploration and high rollout cost.

### 2.2. RL with Internal Rewards

To address sparse rewards in long-horizon agentic RL, recent work leverages model-generated internal feedback to provide denser supervision. Most approaches derive unsupervised rewards from uncertainty, typically measured by policy entropy. However, entropy plays conflicting roles: some methods minimize entropy to encourage confident predictions, while others promote high-entropy exploration by incorporating it into advantage estimation, as in SEED-GRPO (Chen et al., 2025) and related designs. Beyond entropy, DeepConf (Fu et al., 2025b) exploits model-internal confidence to filter low-quality reasoning traces. While these studies show that internal signals can guide exploration, existing methods rely on single-scale heuristics or static reward shaping, lacking principled mechanisms to regulate reasoning across both token and turn levels.

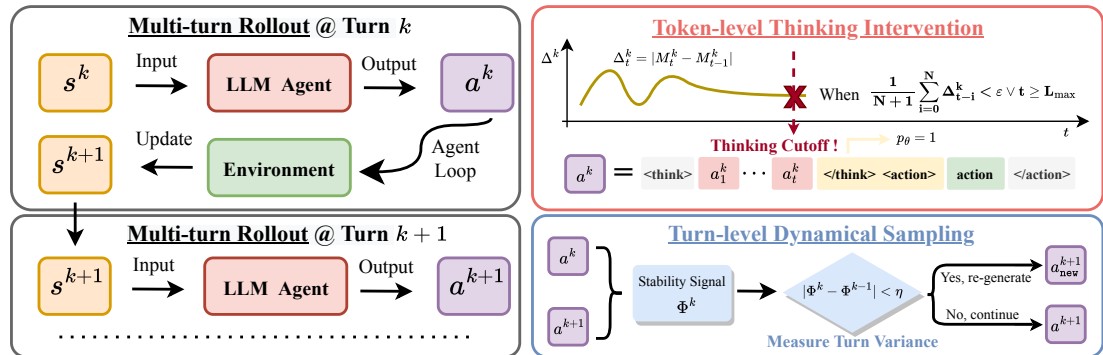

*Figure 2.* Overview of the proposed Uncertainty-Guided Exploration Control at both token and turn levels.

## 3. Preliminaries

We introduce an agentic RL framework that enables an LLM-based agent to interact with external environments and perform multi-turn reasoning to solve complex tasks. Each task begins with a user prompt $q$, which specifies the task description and proceeds over multiple turns $k = \{1, 2, \ldots, K\}$. At each turn $k$, the agent interacts with the environment to obtain an observation represented as the state $\mathbf{s}^k \in \mathcal{S}$, where $\mathcal{S}$ denotes the environment-defined state space. Based on this state, the agent generates an action $\mathbf{a}^k \in \mathcal{V}^n$, $\mathcal{V}^n$ is the action space formed by the LLM tokenizer vocabulary $\mathcal{V}$. Typically, base LLMs fine-tuned with chain-of-thought (CoT) post-training produce both *thinking tokens* $a_c^k$ and *action tokens* $a_o^k$, wrapped in special tags (<think>...</think>, <action>...</action>). Thus, $\mathbf{a}^k$ can be expressed as: $\{a_1^k, a_2^k, \ldots, a_t^k, \ldots, a_T^k\}$, $T$ is the max response length. The agent's behavior is governed by a policy $\pi_\theta(\mathbf{a}^k | \mathbf{s}^k, q)$, which specifies a distribution over possible outputs conditioned on the current state and the initial user prompt. After each action, the environment provides feedback in the form of a scalar reward $r^k \in \mathbb{R}$ and the next state $\mathbf{s}^{k+1}$, unless the maximum number of turns $K$ is reached. Once turn $K$ is completed, a full trajectory is obtained as $\tau = \{(\mathbf{s}^1, \mathbf{a}^1, r^1), (\mathbf{s}^2, \mathbf{a}^2, r^2), \ldots, (\mathbf{s}^K, \mathbf{a}^K, r^K)\}$. In many real-world scenarios, rewards are sparse or delayed, which makes credit assignment particularly challenging given the thousands of tokens generated by LLMs.

## 4. Method

### 4.1. Self-calibrated Uncertainty Signal for Control

**Limitations in typical RL setups.** Token entropy and confidence are commonly used to measure the uncertainty in the token generation distribution. At decoding step $t$ in each turn, the policy LLM model $\pi_\theta$ outputs a categorical probability vector $p_t = \pi_\theta(\cdot | \mathcal{R}_{<t}, x; \mathcal{T}) = \left(p_t^{(1)}, \ldots, p_t^{(V)}\right)$ over a vocabulary of size $V$, where $\mathcal{R}$ is the reasoning trajec-

tory, $x$ is the user prompt, and $\mathcal{T}$ denotes the set of available tools. We quantify uncertainty using Shannon entropy (Lin, 2002) and define token confidence as the negative average log-probability of the top-$j$ tokens at position $t$:

$$H_t = -\sum_{i=1}^{V} p_t^{(i)} \log p_t^{(i)}, \quad C_t = -\frac{1}{j}\sum_{i=1}^{j} \log p_t^{(i)} \quad (1)$$

Low token entropy indicates a sharply peaked distribution and higher certainty, while high confidence likewise reflects greater model certainty.

However, both of them exhibit inherent limitations. Entropy reflects the overall smoothness of the token distribution, but shows limited discriminability at the two extremes, when the distribution is nearly uniform or highly peaked. This limitation becomes particularly pronounced when the vocabulary size is large, such as 152K in Qwen3 (Team, 2025). Since the entropy range scales with $[0, \log V]$, the entropy gap between two highly different predictions, for example, $(1, 0, 0, \ldots)$ and $(0.5, 0.5, 0, \ldots)$, is only $\log 2$. Such a difference is negligible compared with the full entropy scale.

Thus, entropy alone may fail to distinguish between genuinely uncertain predictions and extremely sharp ones. Confidence, in contrast, depends only on the probability of the arg-max token and therefore ignores how the remaining probability mass is distributed. Thus, very different token distributions may yield identical confidence despite different levels of uncertainty (Fu et al., 2020). As shown in Figure 3, both have blind regions:

**Self-calibrated uncertainty signal.** Based on the above analysis, $C_t$ and $H_t$ are complementary in covering both smooth and non-smooth distributions. To obtain a scalar indicator of local distributional stability, we first normalize both entropy and confidence across the decoding trajectory:

$$\tilde{H}_t = \frac{H_t - H_{\min}}{H_{\max} - H_{\min}}, \quad \tilde{C}_t = \frac{C_t - C_{\min}}{C_{\max} - C_{\min}} \quad (2)$$

And we construct a self-calibrated stability signal:

$$M_t = \alpha(\tilde{H}_t) + (1 - \alpha)(1 - \tilde{C}_t), \quad \alpha \in [0, 1] \quad (3)$$

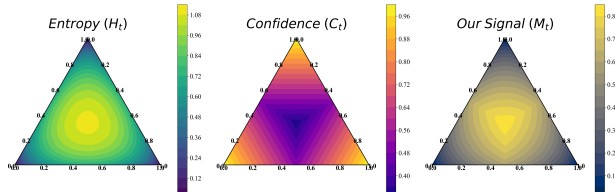

*Figure 3.* Contour of $H_t$ fails to discriminate highly uncertain distributions near uniformity, while $C_t$ ignores variations in tail probabilities. The proposed signal $M_t$ integrates both measures, producing non-degenerate contour geometry that distinguishes distributions sharing identical top-$k$ probability but differing residual mass.

The contour lines of $M_t$ are no longer piecewise-linear and degenerate under the max operator compared with $C_t$. $M_t$ preserves the top-1–driven stratification while introducing curvature within each stratum, enabling it to distinguish distributions with identical $\max(p)$ but different residual mass allocations. Besides, compared with $H_t$, whose contours concentrate around the uniform distribution, $M_t$ produces high-uncertainty regions that align more closely with the existence of a dominant class. Meanwhile, it retains entropy's sensitivity to tail dispersion, yielding uncertainty patterns that better match practical class-confusion behaviors.

### 4.2. Token-Level Thinking Intervention (TTI)

**Motivation.** Reasoning LLMs, inspired by the *aha moment* (Liu et al., 2024), tend to generate elaborate CoT before generating the action. While such reasoning improves decision quality, excessively long internal sequences introduce computational overhead and amplify policy-gradient variance during agent training. Therefore, we continually ask a central question:

> *How can we explicitly and adaptively discard redundant CoT tokens while preserving reasoning performance?*

Our first intuition was to monitor token-level uncertainty signals. As shown in Figure 4 (a), we aggregate trajectories from SOTA baseline and observe that confidence first decreases and then increases, while entropy first rises and then falls. Meanwhile, these most uncertain tokens are precisely the ones the model should generate in a shopping scenario, namely tokens related to product information. More importantly, *the tokens generated after these peak points only slow down exploration efficiency*. Therefore, we propose *TTI* to finely and adaptively terminate reasoning once the predictive distribution exhibits convergence behavior.

**When should we stop?** Higher values of $M_t$ reflect both higher confidence and lower entropy. Therefore, as token generation progresses, the dynamics of $M_t$ serve as a reliable indicator of exploration efficiency. We then monitor the temporal variation of the token $t$ at turn $k$ as $\Delta_t^k = |M_t^k - M_{t-1}^k|$. It starts only after a minimum prefix length $L_{\min}$ is generated to avoid premature truncation. A

*non-hesitation event* is declared when the average variation over a trailing window of size $N$ falls below a tolerance $\varepsilon$:

$$\frac{1}{N+1} \sum_{i=0}^{N} \Delta_{t-i}^k < \varepsilon \qquad (4)$$

We denote the first such time as $t^*$. Intuitively, this marks the point at which the predictive distribution ceases to change meaningfully, indicating that additional reasoning contributes little new information, as shown in Figure 4 (c).

**Why not truncate at the peak?** In Figure 4 (b), the highly exploratory tokens can be broadly categorized into two types. The first type consists of connective or discourse tokens, which are closely related to the model's internal reasoning transitions and often coincide with "aha moments" in reasoning-style generation. The second type corresponds to task-specific tokens, such as product names or attribute descriptors, which carry essential semantic information required for successful task completion. If we directly follow the trend in Figure 4 (a) and truncate each response at the peak point, the truncation would likely occur on task-specific tokens. This would not only fail to improve efficiency, but could also hinder effective exploration by prematurely removing critical semantic content.

Since task-relevant tokens are typically distributed across contiguous segments of the response, the sliding-window aggregation smooths local uncertainty fluctuations and prevents spurious threshold triggers at isolated task tokens. As a result, truncation is activated only when sustained high uncertainty is detected, enabling efficiency gains without obstructing meaningful exploration.

**How to stop?** Once *non-hesitation event* occurs, decoding does not terminate immediately. Instead, at the step $(t^* + 1)$, we explicitly intervene in the model output by forcing the reasoning termination token `</think>` (suppose the token id is `151668`). Let $z_t \in \mathbb{R}^{|\mathcal{V}|}$ denote the pre-softmax logits at step $t$. We overwrite the logits as follows:

$$z_{t^*+1}(v) = \begin{cases} +\infty, & v = 151668, \\ -\infty, & v \neq 151668, \end{cases} \qquad (5)$$

which yields $p_\theta(y_{t^*+1} = \text{</think>} \mid y_{\leq t^*}) = 1$. This operation deterministically terminates the reasoning phase and eliminates stochasticity at the stopping point.

**How is the action generated after stopping?** Following the forced emission of the reasoning terminator, we inject a fixed deterministic token queue $\mathcal{Q} = [\text{</think>}, \text{\textbackslash n}, \text{<action>}]$, starting at step $t^* + 1$. Let $\mathcal{Q}[j]$ denote the $j$-th token in the queue. For $j = \{1, \ldots, |\mathcal{Q}|\}$, we enforce $y_{t^*+j} = \mathcal{Q}[j]$ without sampling from the model distribution. This explicitly delineates the boundary between the reasoning and execution phases, ensuring structured outputs.

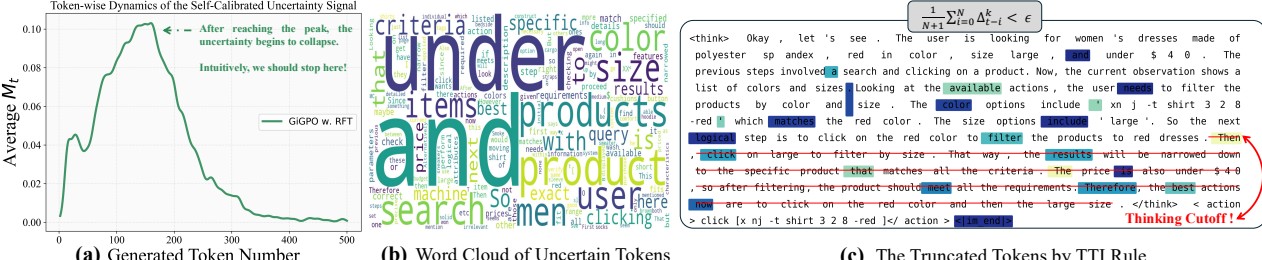

**(a)** Generated Token Number   **(b)** Word Cloud of Uncertain Tokens   **(c)** The Truncated Tokens by TTI Rule

*Figure 4.* (a) Uncertainty dynamics of self-calibrated signal $M_t$ on response length. (b) Word cloud of tokens with the highest uncertainty. (c) Colormap of the uncertainty signal aggregated by the sliding window. When the signal falls below $\epsilon$ (corresponding to the brightest token 'Then'), thinking cutoff is triggered.

**Is there any constraints?** *(1) One-time activation constraint.* To avoid repeated triggering, the stopping mechanism is allowed to activate only once per generation. Let $\mathbb{I}_{\mathrm{stop}} \in \{0,1\}$ be a binary indicator initialized as $0$. The stopping rule is applied only if $\mathbb{I}_{\mathrm{stop}} = 0 \quad \wedge \quad y_t \in \mathcal{V}_{\mathrm{reason}}$, after which we set $\mathbb{I}_{\mathrm{stop}} \leftarrow 1$ and disable further checks. *(2) Global thinking budget.* To guarantee termination, we impose a maximum decoding length $L_{\max}$. If $t = L_{\max}$, we again enforce deterministic termination by overwriting the logits.

**Definition 4.1** (TTI Rule)**.** TTI is triggered if:

$$\frac{1}{N+1}\sum_{i=0}^{N}\Delta_{t-i}^{k} < \varepsilon \quad \vee \quad t \geq L_{\max}, \tag{6}$$

### 4.3. Turn-Level Dynamical Sampling

**Motivation.** Agentic interaction unfolds over multiple turns along a trajectory. At turn level, once the model's perception of the environment stabilizes, it may repeatedly produce semantically similar but failed reasoning traces across turns, leading to redundant interactions and reduced exploration efficiency. A natural inspiration comes from DAPO's dynamical sampling (Yu et al., 2025), which improves sample efficiency by filtering out trivial prompt groups whose accuracy saturates at 0 or 1. However, directly adopting this strategy in multi-turn agentic RL is non-trivial. Unlike single-turn settings where accuracy can be readily computed per prompt group, multi-turn trajectories typically lack dense process rewards and do not admit a well-defined per-turn "accuracy" signal for dynamic filtering. To regularize interaction dynamics at the turn level under this constraint, we introduce a complementary *turn-level dynamical sampling (TDS)* mechanism, which identifies and down-weights redundant turns based on trajectory-level interaction signals.

**Turn-Level control signal.** To measure whether $\mathbf{a}^{k+1}$ is engaging in meaningless exploration compared to $\mathbf{a}^{k}$, we first aggregate all token-level self-calibrated uncertainty signals $M_t^k$ within a single turn. Specifically, the turn-level observation signal is $\Phi^k = \left(\prod_{t=1}^{T} M_t\right)^{\frac{1}{T}}$. We can mon-

itor the temporal variation between consecutive turns as $\Gamma^k = |\Phi^k - \Phi^{k-1}|$. Intuitively, $\Gamma^k$ measures how significantly the model's internal confidence and uncertainty structure have shifted from turn $k-1$ to turn $k$. Large values indicate evolving beliefs or problem-solving states, whereas small values indicate that the agent is repeatedly generating similar, low-information reasoning content.

**When to dynamically sample?** Similarly, we introduce a tolerance threshold $\eta > 0$ controlling the sensitivity of turn-level adaptation. A regeneration event is triggered at turn $k$ when $\Gamma^k < \eta$. That is, if the predictive stability profile of the current turn deviates too much from the previous turn, the turn is deemed insufficiently informative.

**How to dynamically sample?** Specifically, when $\Gamma^k < \eta$, we consider the current turn's action unlikely to efficiently advance the LLM agent toward success. In this case, we discard the current turn's generation and resample a new reasoning trajectory for the same turn. This regeneration process is repeated until a satisfactory trajectory is obtained or a maximum resampling budget is reached. The turn-level control signal is recomputed only after regeneration completes.

**Definition 4.2** (TDS Rule)**.** TDS is defined as follows:

$$\mathbf{a}_{\mathrm{new}}^k \leftarrow \begin{cases} \text{Re-generate}(\mathbf{a}^k), & \text{if } \Gamma^k = |\Phi^k - \Phi^{k-1}| < \eta, \\ \mathbf{a}^k, & \text{otherwise.} \end{cases} \tag{7}$$

where Re-generate$(\cdot)$ denotes a fresh rollout under the same state. This procedure repeats until $\Gamma^k \geq \eta$ or the resampling budget $B_{\max}$ is exhausted. The turn-level control signal $\Phi^k$ is then recomputed after regeneration terminates.

### 4.4. Policy Update

**Memory context window.** Since each task requires multiple interactions with the environment, directly concatenating the entire trajectory $\tau$ for optimization would result in excessively long sequences, which significantly increases computational overhead and memory consumption. Therefore, we adopt a memory context window that includes only the interaction history from the most recent $P$ turns. Concretely, the

current state $\mathbf{s}^K$ contains information from $\mathbf{s}^{K-1}$ to $\mathbf{s}^{K-P}$ and the corresponding actions $\mathbf{a}^{K-1}$ to $\mathbf{a}^{K-P}$, rather than the full trajectory history.

**Credit Assignment.** In practice, reward signal across turns are extremely sparse. To mitigate this issue, a standard approach in multi-turn RL is to introduce a discounted return over turns. Let $\beta \in (0, 1)$ denote the turn-level discount factor. The effective training signal is defined as its discounted return $R(\tau^k) = \sum_{j=k}^{K} \beta^{j-k} r^j$. This formulation propagates supervision from terminal outcomes back to earlier decisions, allowing each action $\mathbf{a}^k$ to be optimized based on its long-term impact on future rewards rather than relying solely on immediate feedback.

**Policy Loss.** T²PO performs hierarchical advantage estimation. Following GRPO, we first group together $G$ full trajectories collected under the same task and identical initial environment states. Then we compute the relative advantage as $A(\tau_i^k) = \frac{R(\tau_i^k) - \mathrm{mean}\left(\{R(\tau_j^k)\}_{j=1}^{G}\right)}{F_{\mathrm{norm}}\left(\{R(\tau_j^k)\}_{j=1}^{G}\right)}$. This captures global performance differences across full interaction trajectories. At the finer scale, we follow GiGPO to compute turn-relative advantage $A^{\mathrm{turn}}$. Finally, we fuse the two signals into a single group-in-group advantage $A'(\mathbf{a}_i^k) = A(\tau_i^k) + \omega \cdot A^{\mathrm{turn}}(\mathbf{a}_i^k)$, which provides outcome and turn-level process credit. The corresponding clipped policy update objective with $\rho_\theta(\mathbf{a}_i^k) = \frac{\pi_\theta(\mathbf{a}_i^k|\mathbf{s}_i^k)}{\pi_{\theta_{\mathrm{old}}}(\mathbf{a}_i^k|\mathbf{s}_i^k)}$ is:

$$\mathcal{J}(\theta) = \mathbb{E}\Big[\min\Big(\rho_\theta(\mathbf{a}_i^k)A'(\mathbf{a}_i^k),\, \mathrm{clip}(\rho_\theta(\mathbf{a}_i^k), 1\pm\epsilon)A'(\mathbf{a}_i^k)\Big)\Big] - \beta\, \mathbb{D}_{\mathrm{KL}}(\pi_\theta\|\pi_{\mathrm{ref}}). \tag{8}$$

# 5. Experiment

## 5.1. Setup

**Tasks.** We evaluate LLM agents on three public available challenging interactive benchmarks including (1) WebShop (Yao et al., 2022), (2) ALFWorld (Shridhar et al., 2020) and (3) Search QA.

WebShop is a web-based interactive environment that tests LLM agents in realistic online shopping scenarios. Agents must navigate a simulated HTML-based shopping website to search for products, browse pages, and complete purchases. The environment contains over 1.1M products and 12k user instructions, providing a rich and diverse action space.

ALFWorld is an embodied environment designed to assess multi-step decision-making, where an agent receives a textual goal and must accomplish it through multi-turn interaction. It comprises 3,827 task instances spanning six categories: Pick & Place, Examine in Light, Clean & Place, Heat & Place, Cool & Place, and Pick Two & Place.

In addition, Search QA includes single-hop QA datasets

like NQ (Kwiatkowski et al., 2019), TriviaQA (Joshi et al., 2017), PopQA (Mallen et al., 2023) and multi-hop QA datasets like HotpotQA (Yang et al., 2018), 2Wiki (Ho et al., 2020), MuSiQue (Trivedi et al., 2021), and Bamboogle (Press et al., 2023). All of the evaluation metrics are shown in Appendix A.1

**Implementation Details.** We use publicly available `Qwen3-4B/8B` models for RFT to regulate behavioral patterns, and initialize training from the `Qwen3-4B/8B-RFT` under three different environment random seeds. In addition to the outcome reward, we incorporate a format penalty to enforce structural compliance. Details are provided in Appendix B. For ALFWorld and WebShop, all RL training methods share identical hyperparameter configurations. The rollout group size for group-based RL methods is set to 8. For Search QA, we follow the experimental settings of Search-R1 (Jin et al., 2025). We adopt E5 as the retriever, set the rollout group size to 5, and limit the maximum number of turns to 4. Notably, we decompose each full trajectory into individual turns for optimization. Our experiments are based on verl (Sheng et al., 2024) RL training framework with the agent loop. All of the experiments are implemented on $8 \times$ NVIDIA H100 GPUs. To ensure a fair comparison, all baselines are initialized from the RFT-based model and employ an identical format penalty to stabilize training.

Additional task details and evaluation metrics are provided in Appendix A. Complete training configurations and hyperparameter details are provided in Appendix A.2. Implementation specifics of RL training techniques, including RFT, format penalty, trajectory decomposition, and policy updates, are described in Appendix B.

**Baseline.** We compare T²PO against a diverse set of strong baselines. (1) Closed-source LLMs: GPT-4o, Gemini-2.5-Pro, and Claude 4, representing sota general-purpose inference model. (2) RL training methods: PPO (Schulman et al., 2017), a standard actor–critic algorithm requiring an auxiliary value model; group-based critic-free methods GRPO (Shao et al., 2024), which perform advantage estimation over trajectory groups; and the SOTA baseline GiGPO (Feng et al., 2025). Additionally, we incorporate effective RL enhancements on top of GiGPO, such as the dynamic sampling proposed in DAPO (Yu et al., 2025). In fact, T²PO is plug-and-play and can be readily integrated with other policy update schemes. We provide additional results based on GSPO (Zheng et al., 2025b) in Appendix D.1.

## 5.2. Main Results

Table 1 presents performance on WebShop and ALFWorld. Direct prompting yields limited success, even for strong proprietary models, while open-source backbones remain substantially weaker under zero-shot inference. (1) Instruction tuning improves reward modeling but fails to produce

*Table 1.* Comparison with different policy optimization methods on WebShop and ALFWorld.

| Method | WebShop | | ALFWorld | | | | | | |
|---|---|---|---|---|---|---|---|---|---|
| | Task Score | Success Rate | Success rate | Pick | Look | Clean | Heat | Cool | Pick2 |
| *Prompting* | | | | | | | | | |
| GPT-4o (Achiam et al., 2023) | 31.8 | 23.7 | 48.0 | 75.3 | 60.8 | 31.2 | 56.7 | 21.6 | 49.8 |
| Gemini-2.5-Pro (Comanici et al., 2025) | 42.5 | 35.9 | 60.3 | 92.8 | 63.3 | 62.1 | 69.0 | 26.6 | 58.7 |
| Claude Sonnet 4 (Anthropic, 2025) | 45.63 | 39.82 | 63.71 | 90.13 | 65.34 | 66.77 | 70.14 | 29.80 | 61.36 |
| Qwen3-32B (Team, 2025) | 25.17 | 5.89 | 25.63 | 63.53 | 18.33 | 18.70 | 24.31 | 10.08 | 10.11 |
| *Instruction Tuning* | | | | | | | | | |
| Qwen3-4B + SFT | 70.91 | 26.56 | 64.06 | 89.29 | 66.67 | 64.12 | 59.26 | 35.71 | 54.54 |
| *RL Training* (Based on `Qwen3-4B-RFT`) | | | | | | | | | |
| PPO (Schulman et al., 2017) | $70.34_{\pm8.63}$ | $61.93_{\pm5.93}$ | $75.39_{\pm3.81}$ | $83.34_{\pm9.47}$ | $75.09_{\pm6.25}$ | $74.50_{\pm7.90}$ | $62.57_{\pm1.56}$ | $84.21_{\pm0.00}$ | $58.33_{\pm7.38}$ |
| GRPO (Shao et al., 2024) | $80.02_{\pm7.94}$ | $68.56_{\pm4.11}$ | $77.35_{\pm0.62}$ | $85.32_{\pm6.77}$ | $64.59_{\pm4.34}$ | $91.16_{\pm0.79}$ | $90.18_{\pm7.15}$ | $73.87_{\pm9.64}$ | $60.20_{\pm5.31}$ |
| GiGPO (Feng et al., 2025) | $86.03_{\pm4.18}$ | $73.83_{\pm3.04}$ | $80.47_{\pm2.43}$ | $87.94_{\pm8.91}$ | $77.31_{\pm8.36}$ | $87.95_{\pm6.87}$ | $86.88_{\pm4.26}$ | $79.09_{\pm4.68}$ | $71.41_{\pm7.08}$ |
| GiGPO + DAPO (Yu et al., 2025) | $86.54_{\pm9.81}$ | $74.02_{\pm8.18}$ | $80.86_{\pm1.37}$ | $89.94_{\pm8.06}$ | $72.08_{\pm0.08}$ | $93.05_{\pm0.43}$ | $79.05_{\pm7.45}$ | $83.08_{\pm7.75}$ | $65.55_{\pm9.12}$ |
| **T²PO (Ours)** | $\mathbf{93.84_{\pm0.22}}$ | $\mathbf{81.64_{\pm0.39}}$ | $\mathbf{90.23_{\pm1.38}}$ | $\mathbf{97.36_{\pm6.94}}$ | $\mathbf{87.77_{\pm4.89}}$ | $\mathbf{98.33_{\pm2.77}}$ | $\mathbf{85.11_{\pm7.64}}$ | $\mathbf{85.84_{\pm2.57}}$ | $\mathbf{80.35_{\pm2.86}}$ |
| *RL Training* (Based on `Qwen3-8B-RFT`) | | | | | | | | | |
| GRPO (Shao et al., 2024) | $79.56_{\pm9.67}$ | $69.47_{\pm8.01}$ | $80.67_{\pm6.36}$ | $90.59_{\pm4.27}$ | $72.12_{\pm7.37}$ | $83.33_{\pm6.12}$ | $70.58_{\pm3.67}$ | $88.91_{\pm5.38}$ | $62.39_{\pm4.72}$ |
| GiGPO (Feng et al., 2025) | $88.76_{\pm5.63}$ | $77.92_{\pm4.87}$ | $85.15_{\pm4.77}$ | $92.10_{\pm9.36}$ | $84.65_{\pm2.84}$ | $89.47_{\pm8.36}$ | $81.25_{\pm7.59}$ | $80.76_{\pm4.02}$ | $75.03_{\pm6.94}$ |
| GiGPO + DAPO (Yu et al., 2025) | $87.95_{\pm4.52}$ | $78.40_{\pm5.12}$ | $89.06_{\pm4.76}$ | $94.73_{\pm3.08}$ | $75.01_{\pm6.37}$ | $98.72_{\pm1.33}$ | $93.75_{\pm3.76}$ | $79.64_{\pm8.25}$ | $75.01_{\pm6.37}$ |
| **T²PO (Ours)** | $\mathbf{91.65_{\pm0.84}}$ | $\mathbf{82.42_{\pm0.61}}$ | $\mathbf{92.41_{\pm1.42}}$ | $\mathbf{99.15_{\pm2.05}}$ | $\mathbf{90.91_{\pm4.37}}$ | $96.67_{\pm3.77}$ | $80.45_{\pm7.79}$ | $\mathbf{90.91_{\pm4.15}}$ | $\mathbf{85.71_{\pm1.46}}$ |

*Table 2.* Performance on search-augmented QA tasks. Models are trained on NQ and HotpotQA with $F_{norm} = $ std. † and ⋆ indicate in-domain and out-of-domain datasets, respectively.

| Method | Type | Single-Hop QA | | | Multi-Hop QA | | | | Avg. |
|---|---|---|---|---|---|---|---|---|---|
| | | NQ† | TriviaQA⋆ | PopQA⋆ | HotpotQA† | 2Wiki⋆ | MuSiQue⋆ | Bamboogle⋆ | |
| **Prompting** | | | | | | | | | |
| GPT-4o (Achiam et al., 2023) | Open-source | - | - | - | - | - | - | - | - |
| Qwen3-32B (Team, 2025) | Open-source | 13.56 | 41.32 | 14.28 | 18.24 | 25.77 | 3.98 | 12.32 | 21.58 |
| **RL Training** (Based on `Qwen2.5-7B-Instruct`) | | | | | | | | | |
| R1-Instruct | Open-source | 21.0 | 44.9 | 17.1 | 20.8 | 27.5 | 6.0 | 19.2 | 22.4 |
| Search-R1 (Jin et al., 2025) | Open-source | 39.3 | 61.0 | 39.7 | 37.0 | 40.1 | 14.6 | 36.8 | 38.5 |
| ZeroSearch (Sun et al., 2025) | Open-source | 43.6 | 61.8 | **51.5** | 34.6 | 35.2 | 18.4 | 27.8 | 39.1 |
| StepSearch (Wang et al., 2025c) | Open-source | - | - | - | 38.6 | 36.6 | **22.6** | 40.0 | - |
| **RL Training** (Based on `Qwen3-4B`) | | | | | | | | | |
| GiGPO (Feng et al., 2025) | Open-source | 44.36 | 63.67 | 46.26 | 39.28 | 39.86 | 13.40 | 70.97 | 52.97 |
| T²PO (Ours) | Open-source | **46.13** | **64.08** | 47.85 | **39.80** | **42.51** | 16.64 | **72.58** | **54.93** |

reliable task completion, highlighting the limitations of imitation learning. (2) RL substantially enhances performance. Among single-turn baselines, GRPO clearly outperforms PPO, confirming the importance of structured policy optimization for stabilizing training. Multi-turn methods further improve success rates, demonstrating the necessity of explicit long-horizon credit assignment. (3) As shown in Table 1, T²PO achieves the best performance across all metrics, reaching success rates of **81.64** on `Qwen3-4B-RFT` and **82.42** on `Qwen3-8B-RFT` on WebShop, and delivering consistent gains of roughly 8–12 points over prior SOTA on ALFWorld. Moreover, T²PO exhibits substantially reduced variance across runs, indicating improved training stability without introducing additional model parameters or environment-specific heuristics.

Table 2 reports results on single-hop and multi-hop QA benchmarks. Our method consistently achieves top performance across single-hop datasets, indicating improved evidence retrieval and grounding. On multi-hop QA, we observe pronounced gains on challenging out-of-domain datasets, particularly on MuSiQue, where our approach

more than doubles prior best performance. Strong results on 2Wiki and Bamboogle further confirm robust multi-step reasoning and generalization.

### 5.3. Ablation on Key Modules

Table 3 presents an ablation analysis to quantify the contribution of each core component in T²PO.

*Table 3.* Ablation study of key modules on WebShop.

| Key Modules | Task Score | Success Rate |
|---|---|---|
| w/o *Rejective Fine-tuning* | 79.28 | 61.32 |
| w/o *Token-level Thinking Intervention* | 81.28 | 73.27 |
| w/o *Turn-level Dynamical Sampling* | 72.40 | 63.67 |
| **T²PO** | **93.84** | **81.64** |

**No RFT cold-start.** RFT on self-distilled data is responsible for filtering malformed or low-quality action during early policy optimization. Without this module, the model exhibits a noticeable degradation in both task score and success rate, indicating that structured rejection fine-tuning

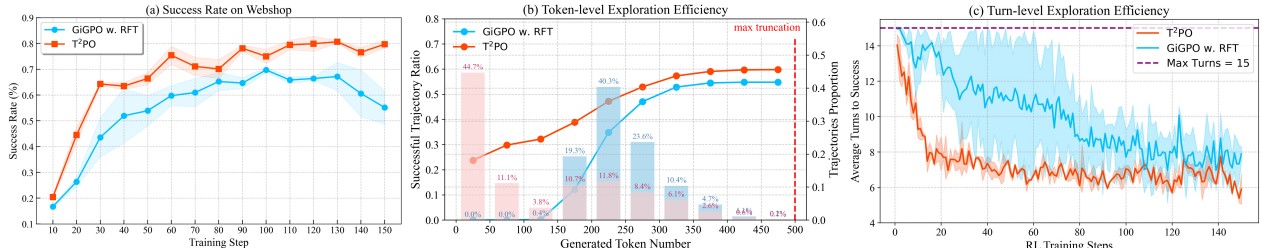

*Figure 5.* We evaluate both task performance and exploration efficiency. (a) shows that T$^2$PO enables performance to steadily improve without collapse on three different env seeds. In (b), the bar chart shows that the distribution of token consumption for successful trajectories generated by T$^2$PO is substantially lower than that of SOTA baseline. Meanwhile, the line plot indicates that the exploration efficiency of T$^2$PO for successful trajectories is consistently higher. (c) further demonstrates at the turn level that T$^2$PO achieves task completion with also $\sim 25\%$ reduced interaction turns during training.

*Table 4.* Ablation of alternative thinking-control methods on Web-Shop with `Qwen3-4B-RFT`.

| Method | Task Score | Success Rate |
|---|---|---|
| Lengthy reward | 77.96 | 65.87 |
| Short CoT cold-start | 82.39 | 71.29 |
| Void turn filtering | 85.17 | 76.20 |
| Hard thinking budget | 84.96 | 79.21 |
| **Ours (*TTI+TDS*)** | **93.84** | **81.64** |

plays a critical role in stabilizing training and preventing error propagation in downstream rollouts.

**Eliminating the *TTI*.** It forces the model to rely on unconstrained reasoning lengths, leading to redundant low-information tokens and inflated trajectory variance. This results in a clear drop in success rate, confirming that adaptive termination based on predictive stability effectively improves exploration efficiency and reduces unnecessary computation without sacrificing reasoning quality.

**Removing the *TDS*.** It is designed to suppress redundant cross-turn reasoning patterns. Without *TDS*, the agent frequently repeats semantically similar reasoning traces across dialogue turns, reducing interaction diversity and limiting effective exploration. Consequently, both task score and success rate deteriorate, demonstrating that turn-level regeneration is essential for maintaining trajectory-level diversity in multi-turn environments.

**Others.** We further investigate the sensitivity of self-calibrated coefficient $\alpha$, tolerance threshold $\epsilon, \eta$, window size $N$ and analyze how varying the maximum response length influences output length and training stability, along with additional ablations on other tasks. The corresponding results are reported in Appendix D.

### 5.4. Ablation on Other Thinking Control Methods

Beyond our hierarchical uncertainty-guided control, we compare T$^2$PO with representative thinking-control strategies with results shown in Table 4, including lengthy reward (Liu et al., 2025b), short CoT cold-start (Cai et al., 2025), hard thinking budget (Comanici et al., 2025) and void turn filtering (Xue et al., 2025). Details of each control method are provided in Appendix C.

(1) The lengthy reward explicitly biases the policy toward shorter generations by penalizing long responses among correct outputs and long incorrect ones, but this global heuristic introduces a rigid preference that does not adapt to task difficulty or per-token predictive stability. As a result, it suppresses both redundant and informative reasoning indiscriminately, yielding only limited performance gains. (2) Short CoT cold-start with data distilled from GPT-4 (Achiam et al., 2023) improves early training stability by initializing the policy with concise teacher demonstrations, yet it does not actively regulate reasoning during RL rollouts; consequently, the model gradually drifts toward repetitive or excessively long reasoning patterns as exploration proceeds. (3) Hard thinking budget imposes a fixed cap on reasoning length. Nevertheless, its static constraint cannot adapt to per-turn uncertainty or task complexity, leading to premature truncation of useful reasoning in difficult cases and insufficient suppression of redundant exploration in simpler ones. (4) Void turn filtering removes trajectories containing invalid or empty actions, preventing trivial degenerate behaviors, but fails to address redundancy among semantically similar valid turns and therefore offers only marginal improvement.

### 5.5. Analysis of Exploration Efficiency

Figure 7 (a) shows that the baseline exhibits early performance improvement but later suffers from instability and partial collapse, whereas T$^2$PO achieves steady monotonic improvement throughout training. This indicates that adaptive thinking regulation stabilizes long-horizon multi-turn policy learning by preventing excessive low-information reasoning from dominating rollouts.

To directly measure token-level exploration efficiency, Figure 7 (b) reports the proportion of successful trajectories as a function of generated token budget. We observe that

T$^2$PO consistently produces a higher fraction of successful reasoning trajectories under the same token budget. In particular, the baseline wastes a substantial portion of tokens on redundant continuation beyond the effective reasoning boundary, while T$^2$PO truncates low-utility reasoning once predictive distributions stabilize. At the turn level, Figure 7 (c) further reports the average number of interaction turns required to complete a task. The baseline agent frequently enters repetitive reasoning loops across turns, leading to longer trajectories and inefficient exploration. In contrast, T$^2$PO detects redundant turn-level reasoning states and triggers regeneration only when necessary, thereby reducing repeated low-information interactions.

### 5.6. Case Study

Details of the trajectory with the interaction between the agent and environment are provided in Appendix G.

## 6. Conclusion

By explicitly regulating reasoning at both token and turn levels using intrinsic signals, T$^2$PO effectively suppresses low-information action and mitigates training collapse without relying on additional reward shaping. Extensive experiments demonstrate that T$^2$PO consistently improves training stability, exploration efficiency, and task performance.

## Acknowledgment

This work was supported by an internship at Amazon.com Inc, NSF 2211557, NSF 2303037, NSF 2312501, NSF 2531008, SRC JUMP 2.0 Center, UCLA CDSC Center, Amazon Research Awards, Snapchat, and Google Gifts.

## Impact Statement

This work advances the understanding of instability in multi-turn reinforcement learning for reasoning-oriented language models. By identifying inefficient exploration as a fundamental cause of training collapse and proposing principled token- and turn-level uncertainty control, our method provides a general framework for stabilizing agentic RL training. We expect this approach to facilitate scalable and reproducible training of interactive LLM agents, enabling broader deployment in complex decision-making applications.

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

# APPENDIX

# A. More Task Details

## A.1. Evaluation Metrics

### A.1.1. WEBSHOP

We adopt six complementary evaluation metrics to comprehensively assess task completion quality. (1) **Task Score** is defined as $10 \times$ avg. reward, measuring the average accumulated reward per episode. (2) **Success Rate** is defined as the proportion of episodes with terminal reward $r = 1$. Notably, an episode may achieve $r = 1$ even if the final selected product does not exactly match the annotated target $y^*$. This is because multiple products may satisfy the same user instruction. For instance, several products can fulfill the request "I want a red shirt", although the instruction was generated from a particular reference item. (3–6) **Title Score**, **reward_type**, **reward_attribute**, and **reward_option** evaluate fine-grained aspects of decision quality, measuring respectively correct product title matching, category consistency, attribute satisfaction, and option-field matching.

Each natural language instruction $u \in \mathcal{U}$ is constructed by human annotators based on a target product $y^*$. It consists of three components: a non-empty attribute set $U_{\text{att}}$, an option field–value set $U_{\text{opt}}$, and a price constraint $u_{\text{price}}$. Formally, $U_{\text{att}} \subseteq Y^*_{\text{att}}$ denotes a subset of the target product attributes, $U_{\text{opt}} \subseteq Y^*_{\text{opt}}$ denotes a subset of its option field–value pairs, and $u_{\text{price}}$ is set higher than the target product price $y^*_{\text{price}}$. This formulation enables lightweight and scalable data collection while preserving realistic user intent. At the end of each episode, the agent receives a terminal reward $r = \mathcal{R}(s_T, a)$, where $a = \texttt{choose[buy]}$, $y$ is the product selected in the final state $s_T$, and $Y_{\text{att}}$, $Y_{\text{opt}}$, and $y_{\text{price}}$ denote the attributes, options, and price of $y$. The reward is defined as:

$$r = r_{\text{type}} \cdot \frac{|U_{\text{att}} \cap Y_{\text{att}}| + |U_{\text{opt}} \cap Y_{\text{opt}}| + \mathbf{1}[y_{\text{price}} \leq u_{\text{price}}]}{|U_{\text{att}}| + |U_{\text{opt}}| + 1}, \tag{9}$$

where the type reward $r_{\text{type}} = \texttt{TextMatch}(\bar{y}, \bar{y}^*)$ penalizes category mismatches between the predicted product $y$ and target product $y^*$. Specifically, $r_{\text{type}}$ assigns a low score when $y$ and $y^*$ share similar attributes or options but belong to different product categories. For example, "butter" and "plant-based meat" may both exhibit attributes such as "cruelty-free" and "non-GMO", yet represent fundamentally different product types.

### A.1.2. ALFWORLD

We follow the standard ALFWorld evaluation protocol (Shridhar et al., 2020). The benchmark contains 3,827 task instances spanning six categories of household activities: *Pick & Place*, *Examine in Light*, *Clean & Place*, *Heat & Place*, *Cool & Place*, and *Pick Two & Place*. Unless otherwise specified, we report overall performance by aggregating results across *all six task categories*. The primary metric is **Success Rate**, defined as the fraction of task instances for which the agent successfully completes the goal. A task is considered successful if the final environment state satisfies all goal conditions specified by the instruction. Overall success rate is computed by averaging over all evaluation episodes pooled from the six task categories.

### A.1.3. SEARCH QA

We evaluate search-augmented reasoning performance using *Exact Match (EM)* as the primary metric. After multi-turn reasoning interleaved with search engine interactions, the model outputs a final answer enclosed by `<answer>` and `</answer>` tokens. The predicted answer $a_{\text{pred}}$ is extracted and compared with the ground-truth answer $a_{\text{gold}}$ via exact string matching:

$$r_\phi(x, y) = \text{EM}(a_{\text{pred}}, a_{\text{gold}}). \tag{10}$$

This EM score serves both as the outcome-based reward for reinforcement learning and as the final evaluation metric during testing. By relying solely on outcome supervision, this metric directly reflects the model's ability to formulate effective search queries, retrieve relevant external knowledge, and integrate retrieved evidence into multi-step reasoning before producing the correct answer. We report EM across seven benchmark datasets covering both general and multi-hop question answering tasks, following standard evaluation protocols.

## A.2. Hyper-parameter Setting

To ensure a fair and controlled comparison across different agentic RL benchmarks, we adopt a unified hyperparameter design principle. All tasks share the same model family, optimization strategy, and rollout framework, while only task-specific parameters are adjusted to match environment characteristics. This standardized configuration eliminates confounding factors from inconsistent training setups, allowing performance differences to be attributed to algorithmic behaviors rather than implementation variance. Accordingly, the hyperparameters in Table 5 are selected to balance training stability, computational efficiency, and reproducibility across tasks.

*Table 5.* Key training hyperparameters for Agentic RL experiments.

| Category | WebShop | ALFWorld | Search |
|---|---|---|---|
| *Model and Environment* | | | |
| Base model | Qwen3-RFT | Qwen3-RFT | Qwen3 |
| Max interaction steps | 15 | 50 | 4 |
| Memory context window | 2 | 2 | 4 |
| Group rollout size | 8 | 8 | 5 |
| Similarity threshold | - | - | 0.9 |
| Max prompt length | 4096 | 2048 | 4096 |
| Max response length | 500 | 500 | 500 |
| Thinking budget | 450 | 450 | 450 |
| *Optimization* | | | |
| Group normalization mode | mean_norm | mean_norm | mean_norm |
| Learning rate | $1 \times 10^{-6}$ | $1 \times 10^{-6}$ | $1 \times 10^{-6}$ |
| Mini-batch size | 128 | 256 | 256 |
| Micro-batch size / GPU | 8 | 32 | 16 |
| KL coefficient | 0.01 | 0.01 | 0.01 |
| Format penalty coefficient | 0.1 | 0.1 | 0.1 |
| *Rollout and Sampling* | | | |
| Rollout engine | vLLM | vLLM | vLLM |
| Rollout mode | synchronous | synchronous | synchronous |
| Tensor parallel size | 1 | 1 | 1 |
| GPU memory utilization | 0.6 | 0.6 | 0.6 |
| Temperature (validation) | 0.6 | 0.6 | 0.6 |
| Top-$p$ (validation) | 0.95 | 0.95 | 0.95 |
| Top-$k$ (validation) | 20 | 20 | 20 |
| Monitoring window size | 20 | 15 | 20 |
| Token-level tolerance $\varepsilon$ | 1e-4 | 1e-4 | 1e-4 |
| Turn-level tolerance $\eta$ | 1e-3 | 1e-3 | 1e-3 |
| *Training Schedule* | | | |
| Train batch size | 128 | 128 | 256 |
| Validation batch size | 128 | 128 | 512 |
| Total epochs | 200 | 150 | 250 |
| *Hardware* | | | |
| GPUs | $8 \times$ NVIDIA H100 | $8 \times$ NVIDIA H100 | $8 \times$ NVIDIA H100 |
| Nodes | 1 | 1 | 1 |

# B. RL Training Techniques

## B.1. Rejective Fine-tuning

A critical challenge in early-stage agentic RL training is the presence of malformed or low-quality action outputs, which introduce substantial noise into trajectory collection and destabilize subsequent policy optimization. To mitigate this issue, we employ a rejective fine-tuning (RFT) stage to initialize the policy with high-quality behavioral priors before RL, without introducing any external supervision or additional knowledge beyond environment feedback. Concretely, we first use the base Qwen3 model to perform multi-turn rollouts in the target environment under the same prompting and tool-calling

format as in RL training to derive $\mathcal{D}_{\text{RFT}} = \{h^k, a^k\}$ with $h^k = \{(s^1, a^1, s^2, a^2, \cdots, s^k)\}$. Each generated trajectory is evaluated by the environment to obtain a scalar task reward. We then retain only trajectories whose final task score exceeds a predefined threshold, discarding low-quality or failed interactions. The remaining high-scoring trajectories are treated as supervised demonstration data, from which we extract state–action pairs and perform one epoch of supervised fine-tuning on the policy model $\pi_\theta$ as follows:

$$\mathcal{L}_{\text{RFT}} = -\mathbb{E}_{(h^k, a^k) \sim \mathcal{D}_{\text{RFT}}}[\log \pi_\theta(a^k | h^k)] \tag{11}$$

This RFT stage equips the agent with a reliable initial policy that produces structurally valid actions and reasonable early reasoning patterns, thereby reducing malformed rollouts and improving training stability in subsequent multi-turn RL.

Notably, we find that RFT provides an effective cold start for agentic RL training. In particular, it significantly strengthens instruction-following ability, leading to more accurate output formatting. Moreover, the success rate of action outputs is substantially improved, as the initial action space is effectively narrowed under RFT initialization. At the same time, we observe that increasing the number of RFT epochs further reduces the RFT training loss. However, excessive RFT begins to degrade the base model's intrinsic reasoning capability, which in turn hinders subsequent RL training. Based on this trade-off, we limit RFT to no more than five epochs in all experiments.

## B.2. Format Penalty

The agent is required to produce actions in a structured format consisting of a reasoning segment and an executable action segment:

```
<think> thinking tokens </think> <action> action tokens </action>.
```

However, during early-stage RL training, LLM agents frequently generate malformed outputs, such as missing tags, duplicated tags, or interleaved natural language, which leads to invalid environment interactions and noisy supervision. To ensure consistent environment interfacing and to construct reliable rejection signals for rejective fine-tuning, we apply a format-constrained action projection operator. Given a batch of raw model outputs $\{a_i^k\}_{i=1}^B$ at turn $k$, we define a strict format validator:

$$\mathcal{V}_{\text{strict}}(a_i^k) = \begin{cases} 1, & a_i^k \text{ matches } \texttt{<think>.*</think><action>.*</action>} \\ 0, & \text{otherwise}, \end{cases} \tag{12}$$

where the match additionally enforces exactly one opening and closing tag for each field and excludes non-target-language characters. If $\mathcal{V}_{\text{strict}}(a_i^k) = 1$, we extract the executable action token from the $\texttt{<action>}$ field and mark the output as format-valid. If the strict constraint fails, we apply a relaxed parser that searches for the first occurrence of an $\texttt{<action>}$ field:

$$\mathcal{V}_{\text{relax}}(a_i^k) = \begin{cases} 1, & \exists \texttt{<action>} \subset a_i^k, \\ 0, & \text{otherwise}. \end{cases} \tag{13}$$

When $\mathcal{V}_{\text{relax}}(a_i^k) = 1$, we still extract the corresponding action token but assign a format-invalid flag. Otherwise, the output is marked as invalid and a fallback placeholder is stored.

To explicitly discourage malformed generations during RL training, we introduce a format-based penalty into the environment reward. Specifically, if an output fails the strict format constraint, i.e., $\mathcal{V}_{\text{strict}}(a_i^k) = 0$, we subtract a fixed penalty from the final task reward:

$$r_i \leftarrow r_i - \lambda_{\text{fmt}}, \quad \text{where } \lambda_{\text{fmt}} = 0.1.$$

This lightweight penalty provides a direct training signal that suppresses malformed thinking–action outputs while avoiding additional external supervision.

## B.3. Trajectory Decomposition

Since we split $\tau = \{(\mathbf{s}^1, \mathbf{a}^1, r^1), (\mathbf{s}^2, \mathbf{a}^2, r^2), \ldots, (\mathbf{s}^K, \mathbf{a}^K, r^K)\}$ into $K$ single turns for policy optimization, it inevitably introduces off-policy staleness. Specifically, in our training pipeline built upon $\texttt{verl}$ (Sheng et al., 2024), rollouts and policy updates are executed in a pipelined fashion: while the learner updates the policy parameters $\theta_\mu$ at training step $\mu$, environment workers may still be generating new trajectories using a previous policy snapshot $\theta_{\mu-\delta}$. As a result, the

*Table 6.* Effect of Off-Policy Staleness under Different Rollout and Group Settings on WebShop.

| Rollout Batch Size $\mathcal{B}_{\texttt{rollout}}$ | Prompt Group Size $n$ | Task Score | Success Rate(%) |
|:---:|:---:|:---:|:---:|
| 8 | 8 | 92.85 | 80.97 |
| 8 | 16 | 93.73 | 81.35 |
| **16** | **8** | **93.84** | **81.64** |

collected turn transitions $(\mathbf{s}^k, \mathbf{a}^k, r^k)$ are not strictly on-policy with respect to the latest parameters, yielding a non-negligible policy lag.

To quantify this effect, suppose the rollout mini-batch size is $\mathcal{B}_{\texttt{rollout}}$, the update micro-batch size is $\mathcal{B}_{\texttt{update}}$, the prompt group size is $n$, and the average turn consumption per trajectory is $\hat{K}_{\max}$ (noting that each trajectory may contain a different number of turns). During training, environment workers continuously generate turn-level transitions, while the learner performs parameter updates in micro-batches. Consequently, before a newly updated policy is broadcast to rollout workers, a number of turn-level samples may already have been generated using stale policy snapshots.

We approximate the expected policy lag, measured in learner update steps, as:

$$\delta \approx \frac{\mathcal{B}_{\texttt{rollout}} \cdot n \cdot \hat{K}_{\max}}{\mathcal{B}_{\texttt{update}}}, \tag{14}$$

which reflects the number of learner updates that can be executed while a batch of rollouts is being collected. Based on this, we define the *staleness ratio* as $\rho_{\text{stale}} = \frac{\delta}{1+\delta}$, which characterizes the fraction of samples generated under outdated policy parameters relative to the total effective update volume. A larger rollout batch size or longer average turn horizon increases $\rho_{\text{stale}}$, whereas larger update micro-batches or higher prompt group parallelism reduce staleness. This ratio therefore provides a concise measure of off-policy deviation induced by turn-level trajectory decomposition in pipelined training.

**Advantages.** It enables scalable multi-turn RL by decoupling environment interaction from policy optimization, thereby significantly improving hardware utilization and training throughput. The decomposition further allows fine-grained control over generation, facilitating dynamic intervention mechanisms such as turn-wise resampling and uncertainty-based stopping. Together, these design choices make it well-suited for long-horizon interactive tasks with large language models.

**Limitations.** The pipelined execution inevitably introduces off-policy staleness, as turn-level samples may be generated under outdated policy snapshots before updated parameters are synchronized across workers. As quantified by the staleness ratio $\rho_{\text{stale}}$, longer interaction horizons and larger rollout batches amplify this effect, potentially increasing importance weight variance and destabilizing policy optimization. To examine whether off-policy staleness has a significant impact on performance, we fix $\mathcal{B}_{\texttt{update}}$ and vary $\mathcal{B}_{\texttt{rollout}}$ and $n$. Table 6 shows that training remains largely stable under these configurations, indicating that off-policy staleness does not substantially degrade performance in practice.

### B.4. Policy Update Details

Our policy update is mainly based on GiGPO (Feng et al., 2025). We also provide the details of turn-relative advantage ($A^{turn}$) computation as follows. Given a rollout group of $G$ trajectories generated for the same task instance, they first enumerate all environment states encountered across all time steps and trajectories, and identify the set of unique states $\mathcal{U}$. Each unique state $\tilde{s} \in \mathcal{U}$ serves as an *anchor state*, around which they gather all occurrences of that state from different trajectories and time steps, forming a turn-level group $G^S(\tilde{s})$. Importantly, this grouping is performed entirely offline via key-based state matching, introducing no additional environment interaction or LLM inference overhead.

For each tuple $(a_i^k, r_i^k)$ in a turn-level group, they compute the discounted return to capture the long-term consequence of the corresponding action. They then normalize these returns within each group to obtain the step relative advantage $A^{turn}$, which measures how well an action performs compared to other actions taken from the same state. This normalization ensures that positive advantages correspond to above-average decisions, while negative values indicate sub-optimal choices under identical state conditions.

# C. More Related Work

Since the core contribution of our method lies in providing **explicit** and **fine-grained** thinking control for multi-turn RL, we also consider a range of established techniques originally proposed for single-turn settings that are closely related in spirit. Therefore, in this section, we present a detailed discussion of lengthy reward (Liu et al., 2025b), short-CoT cold start (Cai et al., 2025), hard thinking budget (Comanici et al., 2025), and void turn filtering (Xue et al., 2025).

## C.1. Lengthy Reward

Over-thinking is also a long-standing challenge in single-turn reinforcement learning for reasoning models. To explicitly regulate excessive reasoning length, Liu et al. (2025b) summarize the existing lengthy reward that incorporates response length into the reward design. Concretely, given a problem $x$ with ground-truth answer $y^*$, suppose a group of responses $\{(y_i, z_i)\}_{i=1}^k$ is sampled, where $z_i$ denotes the reasoning trace and $\text{len}(i)$ is the length of $(y_i, z_i)$. Let $\min\_\text{len} = \min_i \text{len}(i)$ and $\max\_\text{len} = \max_i \text{len}(i)$. If $\max\_\text{len} = \min\_\text{len}$, the length reward is set to zero for all responses since they share identical lengths. Otherwise, the length reward for the $i$-th response is defined as:

$$\text{len\_reward}(i) = \begin{cases} \lambda, & \text{if } r(x, y_i, y^*) = 1, \\ \min(0, \lambda), & \text{if } r(x, y_i, y^*) = 0, \end{cases} \quad \text{where } \lambda = 0.5 - \frac{\text{len}(i) - \min\_\text{len}}{\max\_\text{len} - \min\_\text{len}}. \tag{15}$$

Intuitively, this formulation encourages shorter correct responses while penalizing longer ones among correct outputs, and explicitly penalizes long responses with incorrect answers. The resulting length-based reward is added to the original task reward with a weighting coefficient, providing direct control over the trade-off between reasoning length and task performance.

## C.2. Short-CoT Cold Start

Recent evidence (Cai et al., 2025) suggests that RL in reasoning models does not primarily benefit from memorizing correct solution trajectories, but rather from internalizing structured search behaviors embedded in demonstration traces. This is also why parameter-efficient tuning methods, such as Wang et al. (2023; 2024), can work. In particular, backtracking, where the model explicitly revises earlier decisions, has been identified as a crucial structural prior that enables RL to discover effective multi-step exploration strategies. However, constructing high-quality long reasoning traces with appropriate backtracking is costly and task-dependent.

Motivated by this insight, we adopt short-CoT cold start as a lightweight mechanism for agentic RL. Instead of providing full-length expert trajectories, short SFT initializes the model with concise reasoning patterns from more powerful LLM (*e.g.,* GPT-4o) and basic instruction-following capability, ensuring valid output formatting and a reduced effective action space. This initialization equips the policy with a minimal but consistent interaction protocol, from which RL can reliably amplify and refine latent search behaviors during multi-turn environment interactions.

## C.3. Hard Thinking Budget

Google Gemini 2.5 models (Comanici et al., 2025) provide a dedicated thinking phase designed to improve reasoning and planning in complex tasks. This phase is controlled through a `thinking budget` parameter, which specifies the maximum number of tokens allocated to internal deliberation before the model produces its final response. According to official Gemini and Vertex AI documentation, the thinking process is architecturally separated from the main response generation stage, and users may set an upper bound on its token budget. A special value of $-1$ allows the model to dynamically determine its own budget, while $0$ disables explicit thinking for lightweight variants. Each model family further enforces valid minimum and maximum ranges (*e.g.*, 128–32k tokens for Gemini 2.5 Pro).

## C.4. Void-Turn Filtering.

Void turn filtering (Xue et al., 2025) is a stabilization strategy for thinking control designed to improve the robustness of multi-turn policy optimization. In multi-turn reasoning, the accumulation of low-probability tokens and high sampling stochasticity often produces void turns, *i.e.*, responses that contain neither a valid final answer nor a complete executable structure. Typical void turns manifest as partial code fragments, repetitive text loops, or prematurely terminated outputs caused by early sampling of the end-of-sequence token. Void turn filtering addresses this issue by excluding trajectories

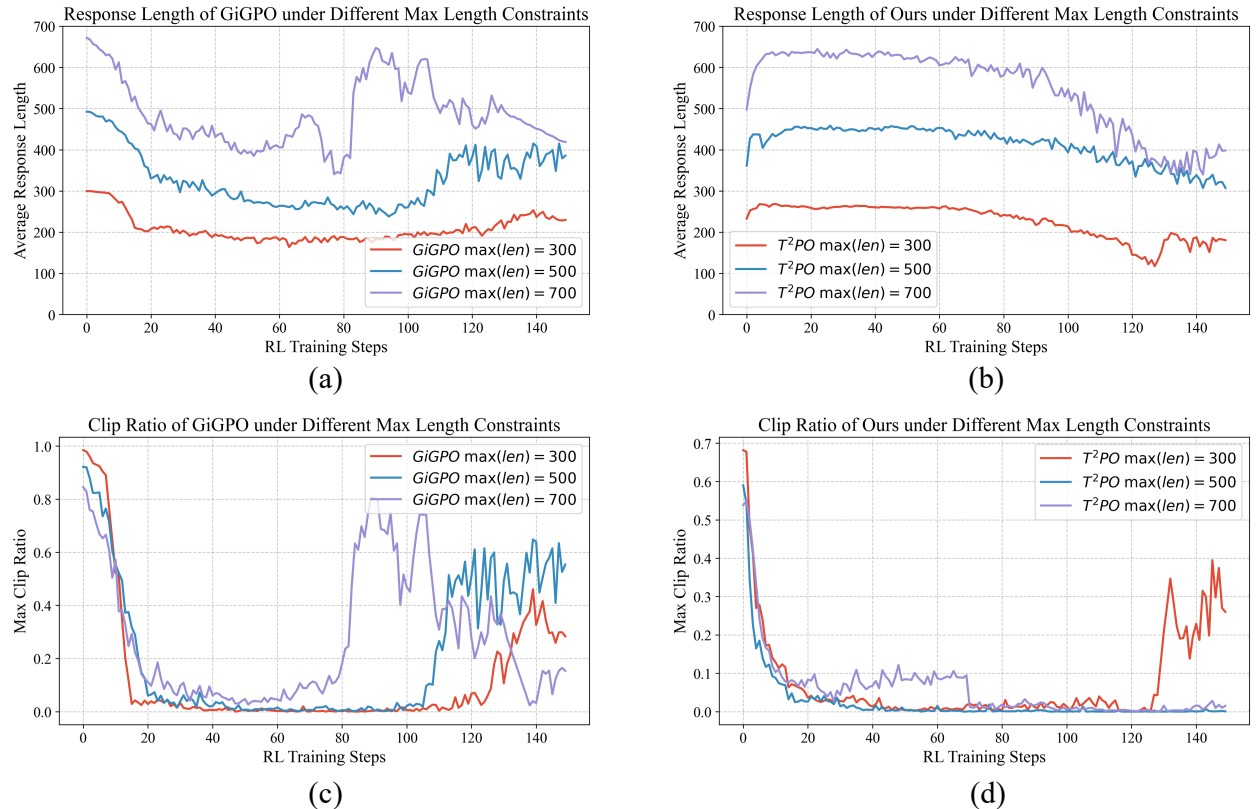

*Figure 6.* (a) illustrates the change in GiGPO's average output length under different maximum response length settings. (b) illustrates the change in T²PO 's average output length under different maximum response length settings. (c) shows the proportion of truncated outputs for GiGPO under different maximum response length settings. (d) shows the proportion of truncated outputs for T²PO under different maximum response length settings.

containing such invalid turns from policy loss computation.

## D. More Experimental Results

### D.1. Ablation Study on other Policy Optimization Algorithm

In fact, our method is plug-and-play and can be readily integrated with other policy update schemes. On WebShop, we further replace the base policy optimization with GSPO (Zheng et al., 2025b). The success rate increases from 85.18 to 91.79 after applying our TTI+TDS, corresponding to a relative improvement of 7.76%.

### D.2. Ablation Study on Token-level Response Length

Figure 6 presents how the policy model's (*i.e.,* Qwen3-4B) output length evolves during training under different pre-specified maximum response lengths (*i.e.*, data.max_response_length in verl (Sheng et al., 2024)). We draw the following conclusions from the observation:

**(1) Longer output length is not always better in multi-turn RL**. From Figure 6 (a) and 6 (b), when the maximum response length increases from 500 to 700, the final model's average output length remains nearly unchanged. This indicates that for interaction-driven environments, excessive token budgets are often unnecessary. Conversely, when the maximum length is too small (*e.g.*, 300), Figure 6 (c) and 6 (d) show that a large fraction of trajectories are still clipped, demonstrating that the token budget is insufficient.

**(2) T²PO achieves higher token efficiency.** Comparing Figure 6 (a) and 6 (b), under the final experimental setting with a 500-token limit, our method produces on average 20% fewer tokens than GiGPO. Meanwhile, Figure 6 (c) and 6 (d) show

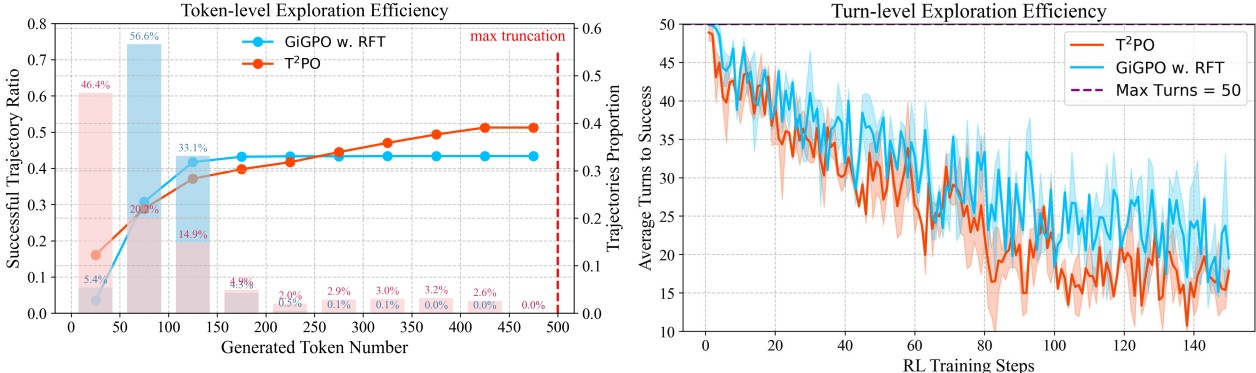

*Figure 7.* Additional efficiency analysis on Alfworld.

that in the last 50 training steps, our method rarely triggers maximum-length clipping, indicating that it avoids generating redundant or uninformative text and substantially mitigates over-thinking.

**(3) $T^2$PO more effectively stimulates meaningful interaction-driven reasoning.** From Figure 6 (a) and 6 (b), our output length gradually increases during the first 20 training steps, reflecting progressively enhanced reasoning depth. In contrast, GiGPO's output length sharply decreases during the first 50 steps, suggesting that it quickly suppresses exploration by discarding many over-thinking trajectories. This observation is further supported by qualitative trajectory case studies.

### D.3. Sensitivity Analysis on $\alpha$

In this section, we conduct a sensitivity analysis on the fusion coefficient $\alpha$ in our self-calibrated uncertainty signal on WebShop, which balances entropy and confidence. We vary $\alpha$ from 0.2 to 0.4, 0.6, and 0.8, and observe that $\alpha = 0.4$ yields the best performance. Therefore, we set $\alpha$ to 0.4.

*Table 7.* Sensitivity analysis of the fusion coefficient $\alpha$ in the self-calibrated uncertainty signal.

| Method | $\alpha = 0.2$ | $\alpha = 0.4$ | $\alpha = 0.6$ | $\alpha = 0.8$ |
|---|---|---|---|---|
| Success Rate | 90.73 | 94.15 | 93.76 | 93.55 |
| Task Score | 79.36 | 82.77 | 81.45 | 80.27 |

### D.4. Efficiency Analysis on Alfworld

We observe a similar phenomenon on ALFWorld. The bar chart shows that the distribution of token consumption for successful trajectories generated by $T^2$PO is substantially lower than that of the SOTA baselines. Meanwhile, the line plot indicates that the exploration efficiency of $T^2$PO on successful trajectories remains consistently higher throughout training. Furthermore, the right figure demonstrates at the turn level that $T^2$PO completes tasks with approximately 16% fewer interaction turns during training.

### D.5. More Results on WebShop

To better understand where performance gains originate, we further report the decomposed reward metrics on WebShop in Table 8. Each reward component evaluates a distinct aspect of task completion, including correct product title identification (*Title Score*), accurate category matching (*reward_type*), attribute fulfillment (*reward_attribute*), and final option selection (*reward_option*).

We observe that prompting-based and instruction-tuned baselines exhibit limited performance on fine-grained reward components, indicating that pure supervised or in-context alignment is insufficient for robust multi-step decision making. Single-turn RL methods (PPO and GRPO) substantially improve all reward dimensions, confirming the benefit of reinforcement learning in aligning long-horizon behaviors. However, multi-turn RL baselines (GiGPO and GiGPO+DAPO) still present imbalanced reward distributions, particularly on *Title Score* and *reward_option*, suggesting that inefficient exploration and unstable credit assignment hinder consistent progress across interaction turns.

---

**Algorithm 1** Token-Level Thinking Intervention (TTI)

---

**Require:** Policy model $\pi_\theta$, minimum prefix length $L_{\min}$, window size $N$, user prompt $q$, stability threshold $\varepsilon$, maximum thinking length $L_{\max}$.

**Ensure:** Generated action sequence $\mathbf{a}^k$ for turn $k$.

1: Initialize token index $t \leftarrow 1$
2: Initialize stop indicator $\mathbb{I}_{\text{stop}} \leftarrow 0$
3: Initialize empty sequence $\mathbf{y} \leftarrow \emptyset$
4: **while** $t \leq L_{\max}$ **do**
5:     Sample next token:

$$y_t \sim \pi_\theta(\cdot \mid y_{\leq t}, q, \mathbf{s}^k)$$

6:     Compute self-calibrated uncertainty $M_t$ using Equation 3
7:     **if** $t > L_{\min}$ **and** $\mathbb{I}_{\text{stop}} = 0$ **then**
8:         Monitor the temporal variation of the token $t$ at turn $k$: $\Delta_t^k = |M_t - M_{t-1}|$
9:         **if** $\frac{1}{N+1}\sum_{i=0}^{N} \Delta_{t-i}^k < \varepsilon$ **then**
10:             Force emit reasoning terminator `</think>`: $z_{t+1}(v) \leftarrow \begin{cases} +\infty, & v = \texttt{</think>} \\ -\infty, & \text{otherwise} \end{cases}$
11:             Append deterministic queue $\mathcal{Q} = [\texttt{</think>}, \texttt{\n}, \texttt{<action>}]$
12:             Set $\mathbb{I}_{\text{stop}} \leftarrow 1$
13:             **break**
14:         **end if**
15:     **end if**
16:     $t \leftarrow t + 1$
17: **end while**
18: **if** $t > L_{\max}$ **then**
19:     **Force emit** reasoning terminator `</think>`
20: **end if**
21: Decode $\mathbf{y}$ into action sequence $\mathbf{a}^k$
22: **return** $\mathbf{a}^k$

---

$T^2PO$ achieves the highest scores on all reward components under both backbones. Notably, gains are most pronounced on *Title Score* and *reward_option*, which require precise information acquisition and decisive action execution. This indicates that our uncertainty-aware optimization effectively suppresses low-information exploration, enabling the policy to focus on high-yield interaction trajectories and produce more reliable fine-grained decisions throughout multi-turn reasoning.

*Table 8.* Reward decomposition on WebShop.

| Method | Title Score | reward_type | reward_attribute | reward_option |
|---|---|---|---|---|
| *Prompting* | | | | |
| Claude Sonnet 4 | 0.3396 | 0.4775 | 0.4263 | 0.4309 |
| Qwen3-4B | 0.1437 | 0.2039 | 0.1756 | 0.0493 |
| Qwen3-32B | 0.1749 | 0.2283 | 0.2003 | 0.0684 |
| *Instruction Tuning* | | | | |
| Qwen3-4B + SFT | 64.58 | 0.8789 | 0.7996 | 0.5826 |
| *RL Training (Qwen3-4B-RFT)* | | | | |
| PPO | $31.65_{\pm 11.45}$ | $33.92_{\pm 9.64}$ | $29.24_{\pm 7.19}$ | $28.76_{\pm 6.99}$ |
| GRPO | $52.92_{\pm 9.13}$ | $51.87_{\pm 11.71}$ | $53.24_{\pm 8.66}$ | $50.49_{\pm 4.89}$ |
| GiGPO | $20.79_{\pm 7.23}$ | $36.20_{\pm 3.28}$ | $27.68_{\pm 6.84}$ | $23.79_{\pm 10.42}$ |
| GiGPO + DAPO | $54.07_{\pm 9.76}$ | $67.20_{\pm 12.45}$ | $54.71_{\pm 8.64}$ | $52.09_{\pm 7.69}$ |
| $T^2PO$ (Ours) | $\mathbf{65.61_{\pm 9.19}}$ | $\mathbf{67.58_{\pm 13.67}}$ | $\mathbf{60.89_{\pm 14.68}}$ | $\mathbf{57.49_{\pm 14.12}}$ |
| *RL Training (Qwen3-8B-RFT)* | | | | |
| GRPO | $53.87_{\pm 6.97}$ | $50.95_{\pm 8.45}$ | $54.67_{\pm 8.53}$ | $50.08_{\pm 9.13}$ |
| GiGPO | $54.98_{\pm 7.45}$ | $54.13_{\pm 4.75}$ | $58.12_{\pm 9.87}$ | $56.28_{\pm 5.67}$ |
| GiGPO + DAPO | $55.88_{\pm 7.01}$ | $68.19_{\pm 7.39}$ | $59.62_{\pm 6.38}$ | $57.44_{\pm 5.37}$ |
| $T^2PO$ (Ours) | $\mathbf{67.14_{\pm 3.73}}$ | $\mathbf{68.98_{\pm 4.18}}$ | $\mathbf{62.77_{\pm 5.04}}$ | $\mathbf{58.33_{\pm 5.34}}$ |

---

**Algorithm 2** Turn-Level Dynamical Sampling (TDS)

---

**Require:** Policy model $\pi_\theta$, user prompt $q$, environment $\mathcal{E}$, turn threshold $\eta$, resampling budget $B_{\max}$, maximum turns $K_{\max}$, target samples $N_{\text{target}}$.

**Ensure:** Collected trajectory set $\mathcal{D}$.

1: Initialize $\mathcal{D} \leftarrow \emptyset$
2: **while** $|\mathcal{D}| < N_{\text{target}}$ **do**
3:     Reset environment: $\mathbf{s}^0 \sim \mathcal{E}.\texttt{reset}()$
4:     Initialize the turn-level observation signal $\Phi^0 \leftarrow \texttt{null}$
5:     Initialize empty trajectory $\tau$
6:     **for** $k = 1$ to $K_{\max}$ **do**
7:         Set resampling counter $b \leftarrow 0$
8:         **repeat**
9:             Generate action $\mathbf{a}^k$ under TTI (Definition 4.1):

$$\mathbf{a}^k \sim \pi_\theta(\cdot \mid \mathbf{s}^k, q),$$

10:             Compute token-level uncertainty $M_t^k$ (Equation 3) and obtain turn-level observation signal:

$$\Phi^k = \Big(\prod_{t=1}^{T_k} M_t^k\Big)^{1/T_k}$$

11:             **if** $k > 1$ **then**
12:                 Monitor temporal variation across turns $\Gamma^k = |\Phi^k - \Phi^{k-1}|$
13:             **else**
14:                 Set $\Gamma^k \leftarrow +\infty$
15:             **end if**
16:             $b \leftarrow b + 1$
17:         **until** $\Gamma^k \geq \eta$ **or** $b \geq B_{\max}$
18:         Parse and execute actions in environment:

$$(\mathbf{s}^{k+1}, r^k) \leftarrow \mathcal{E}.\texttt{step}(\mathbf{a}^k)$$

19:         Store $(\mathbf{s}^k, \mathbf{a}^k, r^k)$ into $\tau$
20:         **if** $\texttt{Is\_all\_done} = \texttt{True}$ **then**
21:             **break**
22:         **end if**
23:     **end for**
24:     Add trajectory $\tau$ to $\mathcal{D}$
25: **end while**
26: **return** $\mathcal{D}$

---

# E. Codebase

Building upon the existing codebase verl (Sheng et al., 2024), our codebase introduces targeted modifications to both the vLLM (Kwon et al., 2023) inference engine and the agent interaction loop, enabling seamless integration with verl while preserving its scalability and modularity. Specifically, we redesign the decoding and rollout pipeline to support fine-grained uncertainty-aware control during generation, while maintaining full compatibility with the step-wise multi-turn training paradigm and memory management mechanisms provided by verl.

In addition, our implementation is framework-agnostic and naturally extends to asynchronous RL training, allowing non-blocking rollout collection and parameter updates under distributed settings. This design ensures that our approach can be directly deployed on top of verl with minimal engineering overhead, while remaining equally applicable to other large-scale async RL infrastructures for LLM agent training.

## F. Algorithm Pseudo Code

Algorithms 1 and 2 summarize the proposed hierarchical exploration control mechanisms. Algorithm 1 presents the Token-Level Thinking Intervention, which dynamically monitors the evolution of the self-calibrated uncertainty signal $M_t$ during token generation. Once the predictive distribution exhibits sustained stabilization according to Definition 4.1, the decoding process is deterministically terminated by overwriting the logits to emit a reasoning terminator token, followed by a fixed structural queue that explicitly separates reasoning and action phases. This design adaptively suppresses low-information continuation while preserving necessary task-relevant reasoning content.

Algorithm 2 describes the Turn-Level Dynamical Sampling procedure, which operates on top of TTI during multi-turn interaction with the environment. For each conversational turn, the turn-level observation signal $\Phi^k$ is computed by aggregating token-level uncertainty across the generated reasoning trajectory. If the variation $\Gamma^k$ between consecutive turns falls below a tolerance threshold, the current turn is deemed insufficiently informative and is regenerated under the same environment state until a sufficiently distinct reasoning trajectory is obtained or a resampling budget is exhausted.

## G. Case Study

**Token-level over-thinking case.** The following figure presents a representative failure case from vanilla GiGPO illustrating how excessive internal reasoning leads to action truncation in long-horizon interactive environments. The agent's state is composed of three structured components: (i) a task specification describing the target product constraints; (ii) a memory context summarizing recent observations and past actions; and (iii) a current observation listing search results together with a discrete set of admissible actions. At each step, the agent must produce a response consisting of a reasoning trace enclosed in <think> ⋯ </think> followed by a single executable command enclosed in <action> ⋯ </action>.

In this example, the reasoning trace grows disproportionately long as the agent attempts to reconcile contradictory attribute constraints (*e.g.*, men's shirt vs. women's fit, fabric requirements, price thresholds, and unavailable color). This induces verbose attribute checking and speculative hypothesis formation, even though none of the listed products match the query. As a result, the generated reasoning exceeds the system's output budget before the closing </think> and <action> tags are produced. The missing termination tags render the response unparsable by the environment, causing an immediate interaction failure despite the correct next step being a simple pagination action.

This case highlights that the action space itself is compact and unambiguous, while the unconstrained reasoning channel becomes the dominant source of failure. It motivates the need for explicit reasoning-length control or early-exit mechanisms to prevent overthinking-induced truncation in multi-turn agentic decision pipelines.

---

**A Failure Case Caused by Over-Thinking**

**System Prompt**

**System**: You're a helpful assistant.
**User**:    You are browsing an online shop.  Based on the instruction, buy a product that close to the production description.  You need to search, read the search results, pick a product, choose the size and color and buy.  You should only choose action from the available actions list provided later.  Example process:  I need a gingko light and 20x20 pillow cover that is hand painted.  First search[gingko light 20x20 pillow cover hand painted], answer format:  <answer>search[blanket with fleece throw]</answer>.  Valid answer is search[<keywords>] or click[<clickable>].

**Turn 6**

---

**State:**

================= User Prompt =================
"You are an expert autonomous agent operating in the WebShop e-commerce environment.Your task is to: Find me wash cold, machine wash men's shirts with polyester heathers, heathers cotton, cotton heather, needle sleeve, classic fit with color: brown, and fit type: women, and size: x-large, and price lower than 40.00 dollars.Prior to this step, you have already taken 5 step(s).

================= Memory Context =================
Below are the most recent 2 observations and the corresponding actions you took: [Observation 4: ''Back to Search' [SEP] 'Page 1 (Total results: 50)' [SEP] 'Next >' [SEP] 'B09P39QN2W' [SEP] 'I Tackled 100 Day Of School Football Boy 100th Day School T-Shirt' [SEP] '$16.99' [SEP] 'B09RVF4JP1' [SEP] 'Once Upon A Time Queen Born In 1982 T Shirts It Was Me Tank Top' [SEP] '$19.99' [SEP] 'B07JVVDJ6L' [SEP] 'Canadian Tuxedo T Shirt – Novelty Denim Tux T-Shirt' [SEP] '$19.99' [SEP] 'B07T3VMGDG' [SEP] 'Tuxedo Cat 4th of July Hat Patriotic Gift Adults Kids Raglan Baseball Tee' [SEP] '$23.99' [SEP] 'B07DKGJR74' [SEP] 'Amazon Brand – Daily Ritual Women's 100% Cotton Oversized Fit V-Neck Pullover Sweater' [SEP] '$20.66 to $29.2' [SEP] 'B08HF132PN' [SEP] 'Flamingo Pink waterbird costume Gift Premium T-Shirt' [SEP] '$19.99' [SEP] 'B09P7H5YK7' [SEP] 'It's In My DNA Palestinian Shirt Arabic Gifts Palestine Flag Tank Top' [SEP] '$19.99' [SEP] 'B07HRFSNL4' [SEP] 'Nautica Men's Solid Crew Neck Short-Sleeve Pocket T-Shirt' [SEP] '$16.05 to $40.98' [SEP] 'B08226NDZW' [SEP] 'Disney Winnie The Pooh Tigger Upside Down Portrait Tank Top' [SEP] '$25.99' [SEP] 'B09Q8RD8YN' [SEP] 'Bungo Stray Anime Dogs Anime Character, Long Sleeve, Sweatshirt, Hoodie, T shirt' [SEP] '$19.99'', Action 4: 'on, but maybe that's'] [Observation 5: ···]

================= Current Observation =================
You are now at step 6 and your current observation is: 'Back to Search' [SEP] 'Page 1 (Total results: 50)' [SEP] 'Next >' [SEP] 'B09P39QN2W' [SEP] 'I Tackled 100 Day Of School Football Boy 100th Day School T-Shirt' [SEP] '$16.99' [SEP] 'B09RVF4JP1' [SEP] 'Once Upon A Time Queen Born In 1982 T Shirts It Was Me Tank Top' [SEP] '$19.99' [SEP] 'B07JVVDJ6L' [SEP] 'Canadian Tuxedo T Shirt – Novelty Denim Tux T-Shirt' [SEP] '$19.99' [SEP] 'B07T3VMGDG' [SEP] 'Tuxedo Cat 4th of July Hat Patriotic Gift Adults Kids Raglan Baseball Tee' [SEP] '$23.99' [SEP] 'B07DKGJR74' [SEP] 'Amazon Brand – Daily Ritual Women's 100% Cotton Oversized Fit V-Neck Pullover Sweater' [SEP] '$20.66 to $29.2' [SEP] 'B08HF132PN' [SEP] 'Flamingo Pink waterbird costume Gift Premium T-Shirt' [SEP] '$19.99' [SEP] 'B09P7H5YK7' [SEP] 'It's In My DNA Palestinian Shirt Arabic Gifts Palestine Flag Tank Top' [SEP] '$19.99' [SEP] 'B07HRFSNL4' [SEP] 'Nautica Men's Solid Crew Neck Short-Sleeve Pocket T-Shirt' [SEP] '$16.05 to $40.98' [SEP] 'B08226NDZW' [SEP] 'Disney Winnie The Pooh Tigger Upside Down Portrait Tank Top' [SEP] '$25.99' [SEP] 'B09Q8RD8YN' [SEP] 'Bungo Stray Anime Dogs Anime Character, Long Sleeve, Sweatshirt, Hoodie, T shirt' [SEP] '$19.99'.Your admissible actions of the current situation are: 'click[back to search]','click[next >]','click[b09p39qn2w]','click[b09rvf4jp1]','click[b07jvvdj6l]','click[b07t3vmgdg]','click[b07dkgjr74]', 'click[b08hf132pn]','click[b09p7h5yk7]','click[b07hrfsnl4]', 'click[b08226ndzw]','click[b09q8rd8yn]'.

================= Instruction =================
Now it's your turn to take one action for the current step.You should first reason step-by-step about the current situation, then think carefully which admissible action best advances the shopping goal. This reasoning process MUST be enclosed within <think> </think> tags. Once you've finished your reasoning, you should choose an admissible action for current step and present it within <action> </action> tags.

**Action:** "<think>Okay, let me try to figure out what to do here. The user is looking for a specific type of men's shirt, but with some conflicting details. The goal is to find wash cold, machine wash men's shirts with polyester heathers, heathers cotton, cotton heather, needle sleeve, classic fit, color brown, fit type women, size x-large, and under $40.Looking at the current observation, all the items listed are either t-shirts, tank tops, or other types of clothing. The first thing I notice is that all the items here are priced under $20 except for the Nautica shirt which has a price range up to $40.98. The user specified a price lower than $40, so that's a bit of a problem because the Nautica shirt's max price is over $40. But maybe there's a specific item in the list that fits the criteria. Wait, the user is looking for men's shirts, but the fit type is women. That seems contradictory. Maybe there's a mistake here. Also, the description mentions "heathers"which are a type of cotton blend, but the items listed here don't seem to match that.

⇑
**Hesitation in Agentic LLMs correlates with the occurrence of an 'aha moment'**

The Nautica shirt is a men's solid crew neck short-sleeve pocket t-shirt, but the fit type is women. That's confusing. The color is brown, but I don't see any brown items here. The price is under $40, but the Nautica shirt is priced up to $40.98, which is over the limit. The other items are all under $20, which is under $40, but none of them seem to match the material or fit type. The user's query includes specific terms like "polyester heathers, "heathers cotton, "cotton heather,"which are types of fabric. The items listed here are either novelty t-shirts, tank tops, or sweaters. None of them mention polyester or cotton heather. The Nautica shirt is cotton, but the fit is women's, which might not fit the user's requirement for men's shirts. Also, the color brown isn't present in any of the listed items. Since none of the items on page 1 seem to match the user's criteria, the next logical step is to check the next page. The previous actions included clicking next >, but the"

⇑
**Overthinking caused truncation: missing </think> and <action> </action> tags**

**Turn-level repeated failure case.** Figure G illustrates a representative failure mode in which the agent becomes trapped in repetitive unsuccessful interaction loops. Specifically, after executing an initial search and clicking a seemingly relevant product, the agent fails to verify whether the item satisfies the required size constraint. Lacking the necessary information to make a correct decision, it returns to the search page and reissues an identical query. This process repeats without meaningful

progress, leading to redundant reasoning, repeated action patterns, and unnecessary token consumption. Such behavior reflects a breakdown in effective exploration, where the agent is unable to adapt its strategy based on newly observed information, ultimately resulting in stalled task completion and inefficient multi-turn interaction.

---

**A Failure Case with Repeated Failure Turns**

**Turn 1**

**Action:** "`<think>` Okay, let's see. The user wants to find men's dress shirts with specific colors (heather grey) and a classic fit, size 3x large, under $30. Current observation is Search, and the admissible actions are to search with a query or click on search. Since the user hasn't done any search yet, the first step is to perform a search with the right criteria. The query should include all the keywords: heather grey, classic fit, 3x large, and price under 30. The search term would be something like heather grey men classic fit 3x large under 30. That should narrow down the results to the relevant products. The next step is to execute this search.`</think>`
`<action>search[heather grey men classic fit 3x large under 30]</action>` "

**Turn 2**

**Action:** "`<think>` Need to look at the first results. The first result is a Nautica t-shirt. It's a heather grey and has a classic fit. The price is $16.05, which is under $30. `</think>`
`<action>click[b07hrfsnl4]</action>` "

**Turn 3**

**Action:** "`<think>` Well, let's see. The user wants men's dress shirts with heather grey, classic fit, and 3x large, under $30. The current observation shows details of that t-shirt. The color here is grey heather, which matches the heather grey the user wants. The size is not mentioned here. The current observation shows the size options. The t-shirt's size is not specified here, but the options are x-small to 6x. The user needs 3x large. The current t-shirt's size is not listed here, but the options are available. The next step is to check the size. I do not have this information and therefore cannot make a correct decision, so we should perform a new retrieval.
`</think>` `<action>click [back to search]</action>` "

**Turn 4**

**Action:** "`<think>` Okay, let's see. The user wants to find men's dress shirts with specific colors (heather grey) and a classic fit, size 3x large, and under $30. The first step is to perform a search with the right criteria. The query should include all the keywords: heather grey, classic fit, 3x large, and price under 30. The search term would be something like heather grey men classic fit 3x large under 30. That should narrow down the results to the relevant products. The next step is to execute this search. `</think>`
`<action>search[heather grey men classic fit 3x large under 30]</action>` "

⇑
**Falling into repeated failures across multiple interaction turns**

---

**T$^2$PO Enables Decisive, Non-Redundant Actions**. The following figure shows a successful WebShop interaction where the agent generates a valid, executable action under the same structured state representation. Concretely, the state is organized into three layers: (i) a task specification that encodes fine-grained attribute constraints (*e.g.*, material, fit, color, size, and price); (ii) a memory context that summarizes the most recent observations and actions, providing short-horizon history for credit assignment and decision continuity; and (iii) a current observation that enumerates the present search-result page together with a closed admissible action set (click targets and navigation operations). This layout aligns the agent's reasoning with the environment's interface: decisions must be grounded in what is currently visible and what can be executed.

Importantly, the generated action content is not redundant. It serves as a compact, outcome-oriented control signal distilled from the multi-constraint reasoning process. Rather than repeating state tokens, the agent uses the memory context to infer that the current result page remains mismatched to the requested attributes, then selects a single navigation operation (`click[< prev]`) that maximally improves the likelihood of reaching a feasible product. This demonstrates strong multi-turn planning and corrective behavior: the agent can leverage interaction history, identify unproductive branches, and execute a minimal yet effective transition. Overall, the case highlights that our method's action generation is both parsable

and decision-effective, reflecting robust reasoning-driven control in long-horizon, constraint-heavy shopping trajectories.

---

**A Successful Case with Valid Action from Our T$^2$PO**

---

**System Prompt**

---

**System**: You're a helpful assistant.
**User**: You are browsing an online shop. Based on the instruction, buy a product that close to the production description. You need to search, read the search results, pick a product, choose the size and color and buy. You should only choose action from the available actions list provided later. Example process: I need a gingko light and 20x20 pillow cover that is hand painted. First search[gingko light 20x20 pillow cover hand painted], answer format: <answer>search[blanket with fleece throw]</answer>. Valid answer is search[<keywords>] or click[<clickable>].

---

**Turn 6**

---

**State:**
================= User Prompt =================
"You are an expert autonomous agent operating in the WebShop e-commerce environment. Your task is to: Find me wash cold, machine wash men's shirts with polyester heathers, heathers cotton, cotton heather, needle sleeve, classic fit with color: brown, and fit type: women, and size: x-large, and price lower than 40.00 dollars. Prior to this step, you have already taken 5 step(s).
================= Memory Context =================
Below are the most recent 2 observations and the corresponding actions you took: [Observation 4: ''Back to Search' [SEP] 'Page 1 (Total results: 50)' [SEP] 'Next >' [SEP] 'B09P39QN2W' [SEP] 'I Tackled 100 Day Of School Football Boy 100th Day School T-Shirt' [SEP] '$16.99' [SEP] 'B09RVF4JP1' [SEP] 'Once Upon A Time Queen Born In 1982 T Shirts It Was Me Tank Top' [SEP] '$19.99' [SEP] 'B07JVVDJ6L' [SEP] 'Canadian Tuxedo T Shirt – Novelty Denim Tux T-Shirt' [SEP] '$19.99' [SEP] 'B07T3VMGDG' [SEP] 'Tuxedo Cat 4th of July Hat Patriotic Gift Adults Kids Raglan Baseball Tee' [SEP] '$23.99' [SEP] 'B07DKGJR74' [SEP] 'Amazon Brand – Daily Ritual Women's 100% Cotton Oversized Fit V-Neck Pullover Sweater' [SEP] '$20.66 to $29.2' [SEP] 'B08HF132PN' [SEP] 'Flamingo Pink waterbird costume Gift Premium T-Shirt' [SEP] '$19.99' [SEP] 'B09P7H5YK7' [SEP] 'It's In My DNA Palestinian Shirt Arabic Gifts Palestine Flag Tank Top' [SEP] '$19.99' [SEP] 'B07HRFSNL4' [SEP] 'Nautica Men's Solid Crew Neck Short-Sleeve Pocket T-Shirt' [SEP] '$16.05 to $40.98' [SEP] 'B08226NDZW' [SEP] 'Disney Winnie The Pooh Tigger Upside Down Portrait Tank Top' [SEP] '$25.99' [SEP] 'B09Q8RD8YN' [SEP] 'Bungo Stray Anime Dogs Anime Character, Long Sleeve, Sweatshirt, Hoodie, T shirt' [SEP] '$19.99'', Action 4: 'click ñext >'']
[Observation 5: ···]
================= Current Observation =================
You are now at step 6 and your current observation is: 'Back to Search' [SEP] 'Page 2 (Total results: 50)' [SEP] '< Prev' [SEP] 'Next >' [SEP] 'B07XPR3R7N' [SEP] 'Marvel Avengers: Endgame Captain America America's Language T-Shirt' [SEP] '$22.99' [SEP] 'B07Q48JVT8' [SEP] 'Marvel Infinity War Thanos Streetwear Poster Graphic Hoodie' [SEP] '$47.99' [SEP] 'B09GL561XH' [SEP] 'Disney Mickey And Friends Happy Birthday Mickey Confetti T-Shirt' [SEP] '$22.99' [SEP] 'B079PH6955' [SEP] 'Need Some Hank & A Drank Country Music T Shirt for Rednecks' [SEP] '$15.49' [SEP] 'B00O30JLDK' [SEP] 'Legendary Whitetails Men's Non-Typical Long Sleeve T-Shirt' [SEP] '$10.52 to $40.5' [SEP] 'B0969G2DH8' [SEP] 'The cool twins just showed up T-Shirt' [SEP] '$17.99' [SEP] 'B085WQKRRJ' [SEP] 'CLANMILUMS Men's Classic Comfort Soft Regular Fit Short Sleeve Henley T-Shirt Tee' [SEP] '$17.99 to $19.99' [SEP] 'B09HX5CD2D' [SEP] 'CSU Cleveland State University Vikings Property Fleece Drawstring Shorts Heather Charcoal' [SEP] '$39.95' [SEP] 'B01MG1LTMS' [SEP] 'Got Snow? Funny Snowmobile Snowboard Skiing Cold Weather Winter Sports Unisex Hooded Sweatshirt' [SEP] '$24.99 to $29.99' [SEP] 'B09R9YCM6R' [SEP] 'XXBR Summer T-shirts for Mens, Soldier Short Sleeve 3D Street Vintage Printed Shirt Slim Fit Muscle Casual Tee Tops' [SEP] '$8.98 to $11.99'. Your admissible actions of the current situation are: ['click[back to search]','click[< prev]','click[next >]','click[b07xpr3r7n]','click[b07q48jvt8]','click[b09gl561xh]','click[b079ph6955]','click[b00o30jldk]','click[b0969g2dh8]','click[b085wqkrrj]','click[b09hx5cd2d]','click[b01mg1ltms]','click[b09r9ycm6r]',].
================= Instruction =================
Now it's your turn to take one action for the current step.You should first reason step-by-step about the current situation, then think carefully which admissible action best advances the shopping goal. This reasoning process MUST be enclosed within <think> </think> tags. Once you've finished your reasoning, you should choose an admissible action for current step and present it within <action> </action> tags.assistant

---

**Action:** "<think>Okay, let's tackle this step by step. The user is looking for a specific type of men's shirt, but the search results seem to be going in the wrong direction. The initial query was for wash cold, machine wash men's shirts with polyester heathers, heathers cotton, cotton heather, needle sleeve, classic fit, color brown, women's fit, size x-large, under $40. Looking at the current observation, the search results are on page 2, and the items listed don't match the criteria. For example, the first item is a Marvel Avengers T-shirt priced at $22.99, which is way too pricey and not the right type. The next items are things like a Thanos hoodie ($47.99), a Disney Mickey T-shirt ($22.99), a country music T-shirt ($15.49), and others. None of these seem to be men's shirts with the specified materials or fit. The price is under $40, but some items are over that, like the Thanos hoodie. The materials mentioned (polyester heather, cotton heather) aren't listed here either.Since the current page (page 2) doesn't have any items matching the criteria, the next logical step is to go back to the previous page (page 1) to check if there's a better result there. Alternatively, if page 1 also doesn't have the right items, the user might need to adjust the search terms, but since the user hasn't done that yet, it's better to check the previous page first. So the best action here is 'click[< prev]' to go back to page 1 and look again.</think><action>click[< prev]</action>"

⇑

**We can successfully parse the action!**

