# OpenReview forum: "T$^2$PO: Uncertainty-Guided Exploration Control for Stable Multi-Turn Agentic Reinforcement Learning"
_ICML.cc/2026/Conference — ICML 2026 spotlight_

### Official Review · Reviewer_LbQJ · 2026-03-09

**Soundness:** 3
**Presentation:** 3
**Significance:** 3
**Originality:** 3
**Overall Recommendation:** 4
**Confidence:** 3

**Summary:**

This paper introduces T^2PO to prevent LLM agents from crashing during multi-turn RL training. It monitors an uncertainty signal in real time to keep the agent on track. At the token level, it cuts off the agent's reasoning the second it stops generating useful information. At the turn level, if an interaction step shows no real progress, the system simply resamples. Tests on benchmarks like WebShop and ALFWorld show this approach effectively stops training collapse and boosts overall success rates.

**Compliance With Llm Reviewing Policy:**

Affirmed.

**Final Justification:**

The rebuttal has addressed my main concerns.

**Key Questions For Authors:**

See Weaknesses

**Limitations:**

The authors did not discuss the limitations.

**Strengths And Weaknesses:**

# Strengths

1. T^2PO uses a real-time, self-calibrated uncertainty signal to detect and hard-stop "overthinking" during token generation.
2. T^2PO tackles exploration inefficiencies at both the micro level (token-level thinking intervention) and macro level (turn-level dynamical sampling)
3. The framework is algorithm-agnostic and can easily integrate with existing policy update schemes like PPO, GRPO, and GiGPO.

# Weaknesses
1. In RL, highly uncertain decision-making points often require *more* exploration. By prematurely cutting off these uncertain traces, T^2PO might restrict the agent from discovering diverse and potentially optimal possibilities.
2. Forcing the agent to favor only deterministic tokens or turns could lead to homogeneous behaviors or cause the agent's policy entropy to collapse. The paper currently lacks sufficient analysis on this issue, such as providing entropy evolution curves.
3. The turn-level dynamical sampling might mistakenly interrupt an agent if a complex task legitimately requires executing highly similar exploratory actions across consecutive steps.

---

> ### Author Rebuttal · Authors · 2026-03-31
>
> > Q1: In RL, highly uncertain decision-making points often require *more* exploration. By prematurely cutting off these uncertain traces, T^2PO might restrict the agent from discovering diverse and potentially optimal possibilities.
> >
>
> A1: Thank you for raising this concern. T$^2$PO does not truncate trajectories merely because uncertainty is high; rather, it is designed to filter out redundant exploration in the reasoning process. The trigger is adaptive and change-based: intervention occurs only when the uncertainty dynamics have largely stabilized, indicating that continued exploration is no longer generating meaningfully new information and is more likely to add rollout noise than useful diversity. In other words, the method is designed to preserve productive exploration rather than suppress it.
>
> Moreover, truncation in our framework is not an irreversible termination of reasoning. After cutoff, the agent can still resume and continue reasoning from the retained prefix, so the procedure does not prevent the model from exploring alternative promising directions. Instead, it removes low-information continuation while preserving the ability to re-enter exploration when needed. For this reason, we do not view TTI as prematurely eliminating diverse possibilities, but as reallocating exploration budget away from redundant traces toward more informative ones.
>
> Empirically, if truncation were indeed harming useful exploration, we would expect degraded final performance or reduced robustness. In contrast, our results show the opposite: the full method consistently improves performance and stability, suggesting that the adaptive cutoff is not restricting meaningful exploration, but filtering out unproductive continuation.
>
> > Q2: Forcing the agent to favor only deterministic tokens or turns could lead to homogeneous behaviors or cause the agent's policy entropy to collapse. The paper currently lacks sufficient analysis on this issue, such as providing entropy evolution curves.
> >
>
> A2: Thank you for the comment. To directly address your concern, we show the entropy evolution curve during training on both ALFWorld and WebShop at the following anonymous link: https://anonymous.4open.science/r/T2PO-ICML-3C21/entropy_curve.md.
>
> The resulting curves do not show entropy collapse. On ALFWorld, the entropy remains stable in a clearly non-zero range throughout training, with persistent local fluctuations even at later stages. On WebShop, the entropy decreases from an initially high level as training becomes more structured, but it does not collapse toward zero; instead, it stabilizes at a moderate level and continues to vary substantially across training steps. If T²PO were merely suppressing exploration, we would expect entropy to rapidly decrease and remain near zero, which is not what we observe. These results suggest that T²PO does not eliminate exploration, but rather improves its quality.
>
> > Q3: The turn-level dynamical sampling might mistakenly interrupt an agent if a complex task legitimately requires executing highly similar exploratory actions across consecutive steps.
> >
>
> A3: Thank you for the comment. We clarify that TDS does not suppress **similar** **actions** themselves. It only intervenes when two consecutive turns exhibit highly **similar** **uncertainty dynamics**, meaning that the later turn is no longer materially updating the model’s internal state. Thus, repeated actions are not penalized as long as they remain informative. For example, in WebShop, later turns often repeatedly take the form of ``click xxx”, but TDS does not suppress them if these actions continue to reduce uncertainty and advance the interaction. TDS is mainly introduced for multi-turn GRPO-style learning, where supervision exists only at the trajectory level rather than the intermediate-turn level. In this setting, the key requirement is to maintain diversity among trajectories within the same sampled group, since step-wise rewards are unavailable to encourage useful exploration diversity. When group trajectories repeatedly follow nearly identical turn-level exploration patterns, the relative learning signal becomes much less informative. TDS mitigates this issue by promoting group-level diversity, while a maximum retry budget ensures that regeneration remains a conservative and bounded intervention.

---

> > ### Author Rebuttal · Reviewer_LbQJ · 2026-04-03
> >
> > Thanks for the responses.

---

> > > ### Author Response · Authors · 2026-04-03
> > >
> > > Thanks for your suggestions. We are gratified that our responses have satisfactorily addressed your concerns, and we sincerely appreciate your careful consideration and constructive feedback throughout the review process.

---

### Official Review · Reviewer_E63u · 2026-03-10

**Soundness:** 3
**Presentation:** 4
**Significance:** 3
**Originality:** 3
**Overall Recommendation:** 5
**Confidence:** 4

**Summary:**

This paper addresses the issue of training instability and collapse in multi-turn reinforcement learning for LLM agents.
Introduce Token- and Turn-level Policy Optimization, $T^2PO$, a framework that uses a novel, self-calibrated uncertainty signal to adaptively control exploration.

**Compliance With Llm Reviewing Policy:**

Affirmed.

**Key Questions For Authors:**

- Mentioned in the weaknesses, RFT is crucial for the performance of the method. More analysis is needed to understand the contribution of RFT. Without RFT, how does the baseline perform on WebShop and ALFWorld?
- How does $T^2PO$ perform on smaller models like Qwen3-1.7B? Do the optimal values of $\epsilon$ and $\eta$ change with model size? Are there any guidelines for choosing these hyperparameters?
- Does the size of this window (P) impact the stability of the $M_t$, or is the uncertainty calculation purely local to the current turn's logits?

**Limitations:**

yes

**Strengths And Weaknesses:**

Strengths
- This paper develop a novel self-calibrated signal ($M_t$) by fusing entropy and confidence.
- The method prevents the collapse often seen in vanilla RL pipelines. Empirical results show substantially reduced variance across different seeds.
- The writing is clear and the paper is well-structured.

Weaknesses
- The framework relies on RFT to initialize the policy. Ablation studies show a performance drop without RFT, suggesting $T^2PO$ might struggle to stabilize training without RFT?

---

> ### Author Rebuttal · Authors · 2026-03-31
>
> > Q1: Role of RFT.
> >
>
> A1: Thanks for your comment. RFT stabilizes training and provides a fair testbed. All baselines are trained from the Qwen3 model initialized by RFT, which helps avoid training instability caused by format following errors.
>
> Notably, RFT is **orthogonal to our core method design**. Our method-specific contribution is the exploration-control mechanism itself, namely token-level intervention and turn-level dynamic resampling based on uncertainty dynamics. Under the same shared infrastructure, removing these exploration-control components leads to clear degradation, which indicates that the gain is not explained by the common training recipe alone but is driven by the proposed exploration-control design.
>
> > Q2: Contribution of RFT.
> >
>
> A2: Thank you for the comment. We agree that the contribution of RFT should be clarified more explicitly. The role of RFT is to use high-quality trajectories to narrow the effective action space, so that the model can more reliably produce valid actions in the required format, i.e., \<action\> output \</action\>. In this sense, RFT mainly serves as a training stabilizer: it reduces failures caused by formatting errors, mitigates early collapse, and improves learning efficiency, rather than constituting the core innovation of our method.
>
> This is also why we apply RFT to all methods in Table 1 as part of a fair and stable training testbed. Our goal is to keep the initialization and formatting stability conditions fixed across methods, so that the comparison focuses on the contribution of the exploration-control mechanism rather than on avoidable optimization artifacts.
>
> On WebShop, we did run early experiments without RFT for our method. In two runs without RFT, the success rates were 23.56 and 73.42, compared with an average of 81.64 with RFT. In the first run, the model frequently failed to follow the required action format, could not produce valid actions, and eventually collapsed during training. In the second run, the model overcame the formatting issue, but the learning process was much slower; after extending training to 300 steps, it reached 81.33, which is close to the final performance with RFT. These results suggest that RFT mainly **improves robustness and sample efficiency**. We will add the corresponding clarification in the revision.
>
> > Q3: Performances on Qwen3-1.7B. Do the optimal values of and change with model size? Are there any guidelines for choosing these hyperparameters?
> >
>
> A3: Thanks for the comment. We add the experimental results of Qwen3-1.7B on WebShop under the same parameter settings, and we can see that it still substantially outperforms the baseline. We may therefore say that, as long as the base model belongs to the same model family, the parameters generally do not need to be changed much, because the distribution of the policy model is basically consistent. In fact, the specific settings we used for the three tasks in Table 5 are also largely similar.
>
> | Method | Task Score | Success Rate |
> | --- | --- | --- |
> | GiGPO | 80.21 | 70.86 |
> | Ours | 87.39 | 78.44 |
>
> Our guidance is that as long as the reward curve continues to rise steadily, there is no need to tune the parameters. If performance degrades severely and the probability of trajectory truncation increases, then the strength of stop-thinking can be increased.
>
> > Q4: Does the size of this window (P) impact the stability of the M_t, or is the uncertainty calculation purely local to the current turn's logits?
> >
>
> A4: Thanks for the comment. The uncertainty score $M_t$ itself is computed locally from the current token logits. It does not depend on the window size P. The role of P is only in the stabilization test built on top of $M_t$: we use a sliding window to smooth short-term fluctuations in the sequence of uncertainty changes, so that the decision is not overly influenced by anomalous tokens. We have also added an ablation study on P, which shows that when P is very small, it may be affected by fluctuations from anomalous tokens. Once P exceeds 10, the behavior becomes very stable.
>
> | Window Size | 5 | 10 | 20 | 30 |
> | --- | --- | --- | --- | --- |
> | Task Score | 90.89 | 93.04 | 93.84 | 92.58 |
> | Success Rate | 77.50 | 80.85 | 81.64 | 80.03 |

---

> > ### Author Rebuttal · Reviewer_E63u · 2026-04-04
> >
> > Thanks for the detailed rebuttal, and I will keep my positive score.

---

> > > ### Author Response · Authors · 2026-04-05
> > >
> > > Thank you for your positive score! We are gratified that our responses have satisfactorily addressed your concerns, and we sincerely appreciate your careful consideration and constructive feedback throughout the review process.

---

### Official Review · Reviewer_YnSQ · 2026-03-12

**Soundness:** 3
**Presentation:** 2
**Significance:** 2
**Originality:** 2
**Overall Recommendation:** 4
**Confidence:** 3

**Summary:**

This paper studies unstable training in multi-turn agent RL. The authors argue that one main cause is poor exploration, which they describe as hesitation. They propose T2PO, a method that uses token-level uncertainty to stop unhelpful long reasoning and turn-level uncertainty to regenerate weak turns. The results on WebShop, ALFWorld, and Search QA are strong. The method improves both average performance and training stability compared with baselines.

**Compliance With Llm Reviewing Policy:**

Affirmed.

**Final Justification:**

The rebuttal addressed my main concern and made the contribution more convincing, so I raise my recommendation to weak accept.

**Key Questions For Authors:**

How much of the improvement remains if you keep the same training setup and add only the exploration-control component?

How sensitive is the method to the thresholds, window size, and resampling budget across different tasks?

Can you show examples where the token-level stopping rule cuts off reasoning that would actually have been useful?

Why is your uncertainty score better than simpler alternatives?

**Limitations:**

yes

**Strengths And Weaknesses:**

The main idea is easy to understand and the distinction between token-level control and turn-level control is reasonable. The empirical results look strong across three benchmarks. The ablation studies are useful.

My main concern is that the method still seems quite heuristic. The uncertainty score is built from entropy and confidence, but the paper does not clearly explain why this is the right combination. The stopping rule and regeneration rule also rely on several thresholds and budgets. Because of this, the method may be less general than the paper suggests.

It is also difficult to tell which part really drives the improvement. The full system includes many components: cold-start data, a memory window, a format penalty, return design, and the policy optimization recipe. Because of this, the paper does not fully show that better exploration control is the main reason for the gain.

The writing is mostly clear, but some design choices are supported more by intuition than by analysis. The overall structure is fine, but the paper should separate the core method from the additional training tricks more clearly.

---

> ### Author Rebuttal · Authors · 2026-03-31
>
> > Q1: Why is the uncertainty score the right combination?
> >
>
> A1: Thanks for the comment. We agree to claim our self-calibrated signal more clearly. Our choice is not meant to be an ad-hoc feature combination, rather, it is motivated by the complementary failure modes of the two signals discussed in Section 4.1.
>
> Entropy captures the overall smoothness of the token distribution, but has limited discriminability at the two extremes, especially under large vocabularies. Confidence, by contrast, is driven mainly by the top-ranked probability and ignores how the remaining probability mass is distributed. As a result, entropy alone may fail to distinguish genuinely uncertain predictions from extremely sharp ones, while confidence alone may assign similar values to distributions with very different uncertainty structures.
>
> Thus, the self-calibrated signal is introduced precisely to address these complementary blind spots. It preserves the top-1-driven stratification from confidence while retaining entropy’s sensitivity to tail dispersion. This yields a non-degenerate uncertainty geometry that can distinguish distributions sharing similar top-k probabilities but differing in their residual mass allocation. Therefore, the motivation is not to arbitrarily combine two common uncertainty measures, but to construct a signal that avoids the degeneracies of using either entropy or confidence alone.
>
> > Q2: It may be less general.
> >
>
> A2: Thanks for the comment. Both the TTI and TDS rule are derived from the same **task-agnostic** **principle**. Intervention is triggered only when the uncertainty dynamics indicate low-information or low-progress exploration. The thresholds and budgets therefore specify the operating point of the controller, not benchmark-specific behavior. This is qualitatively different from hand-crafted heuristics that depend on environment semantics or manually designed action templates.
>
> Empirically, the method is not validated on a single environment or a single backbone. We evaluate the same exploration-control mechanism across **3 different tasks and 2 models per task**, and observe consistent gains. If the improvement mainly came from fragile threshold engineering, one would expect a very narrow operating regime and unstable behavior across settings.
>
> > Q4: Which part really drives the improvement?
> >
>
> A4: Thanks for the comment. The components you mentioned are not unique to our method. They are applied to all baselines in Table 1 as part of a common training testbed for stabilizing Qwen3 training.
>
> Their purpose is to reduce confounding factors such as formatting failures and generic optimization instability, so that the comparison isolates the algorithmic difference of interest rather than incidental training artifacts. In that sense, these components are **orthogonal to our core method design**. Our method-specific contribution is the exploration-control mechanism itself.
>
> > Q5: How much is your own improvement?
> >
>
> A5: Thanks for the comment. All comparisons are conducted on the unified and fair testbed. Therefore, all observed improvements can be attributed solely to the exploration-control component.
>
> > Q6: How sensitive is the method?
> >
>
> A6: Thanks. Please refer to A3, in response to Reviewer **A1UR**, for the sensitivity analysis on the tolerance threshold, and to A4, in response to Reviewer **E63u**, for the analysis of window size.
>
> > Q7: Examples of TTI.
> >
>
> A7: Thanks. Figure 4(c) provides a detailed visualization of what the truncated trajectories look like. Detailed case study is in Appendix G.

---

> > ### Author Rebuttal · Reviewer_YnSQ · 2026-04-03
> >
> > Thank for the response. I would like to see a component-isolation ablation since you didn’t provide one as I requested in my review: run compute-matched comparisons on WebShop (i) no exploration control vs. (ii) full TTI+TDS, holding the entire training recipe fixed and matching either total sampled tokens or environment steps (and ideally reporting both). Please also report TDS usage statistics (regen rate, average retries, and how often the retry cap is reached). This would directly test the causal claim in the rebuttal under a controlled budget.

---

> > > ### Author Response · Authors · 2026-04-05
> > >
> > > Thank you very much for your further detailed guidance in clarifying your concern. In fact, all of our baselines include two budget control metrics: the maximum response tokens per step and the maximum number of turns. In Table 6, we also present an ablation study under different budget settings, and in Table 4 we provide an ablation on the hard thinking budget as well.
> > >
> > > However, because our method introduces an additional control mechanism, the overall token consumption and number of interaction turns vary dynamically, rather than being strictly equal to those of the baselines. Therefore, this is not yet a strict compute-matched causal test. Following your suggestion, we would be very happy to **add an additional experiment** that matches either the total sampled tokens or the environment steps, and to further report the detailed TDS metrics.
> > >
> > > Specifically, we rerun the Full TTI+TDS experiment on WebShop to obtain a step-wise reference training budget, and then let the No Exploration Control baseline run until it reaches the same cumulative budget. For a fair comparison:
> > >
> > > - All experiments use the same backbone, the same RFT initialization, the same format penalty, the same GiGPO update, the same rollout group size, the same maximum interaction steps, the same maximum response length, and the same training batch configuration.
> > > - We also include the extra tokens consumed by regeneration in the token consumption budget.
> > > - Because the budget changes dynamically during training, we record the average token and turn budget at each training step under the same random seed and use them as the reference budget for the No Exploration Control baseline.
> > > - Training is stopped once the same number of tokens has been consumed or the same number of interaction turns has been reached.
> > >
> > > Here are the experimental results:
> > >
> > > | Method | Score | Success |
> > > | --- | --- | --- |
> > > | No Exploration Control | 86.47 | 74.59 |
> > > | Full TTI+TDS | 93.15 | 81.96 |
> > >
> > > This further demonstrates that our fine-grained control strategy can boost agentic RL training under the same budget while improving exploration efficiency throughout the training process.
> > >
> > > We also additionally report the final TDS runtime statistics across all samples during this experiment. The average **regeneration rate** (the proportion of turns that triggered at least one regenerate operation out of all turns) is 16.7%. The **average number of retries** (the average number of additional resampling attempts on turns where regeneration is triggered) is 1.5. And the **retry-cap-reached frequency** (the proportion of regenerate-triggered turns whose number of resampling attempts exceeded 3, among all regenerate-triggered turns) is 11.3%.

---

### Official Review · Reviewer_A1UR · 2026-03-13

**Soundness:** 3
**Presentation:** 3
**Significance:** 3
**Originality:** 3
**Overall Recommendation:** 4
**Confidence:** 4

**Summary:**

This paper proposes T^2PO, a framework for controlling exploration based on uncertainty signals like entropy and confidence at token-level and turn-level. At token-level, rollouts are truncated if they are low-information, and at the turn-level, rollouts are re-generated if there's not enough difference from previous rounds. Experimental results on Webshop, alfworld, searchQA show better performance comparing to baselines.

**Compliance With Llm Reviewing Policy:**

Affirmed.

**Final Justification:**

The response has addressed my concerns.

**Key Questions For Authors:**

* Can the authors provide the sensitivity analysis on the parameters that are used in TTI rule and TDS rule?
* The TDS rule seems to be very fragile in the sense that it can encourage superficial variation in the reasoning text with little novelty. Has the authors observed this and taken any measures to prevent such superficial variations?
* Can the authors compare with prior work where similar token-level thinking cutoff is adopted like [1]?

[1]: Fu, Yichao, et al. "Deep think with confidence." arXiv preprint arXiv:2508.15260 (2025).

**Limitations:**

* The authors should report wall-clock time or total compute budget for the experiments.

**Strengths And Weaknesses:**

Strengths:
* Paper is generally well-written and easy to follow, with clear figures to illustrate the core idea.
* The paper addresses an importantly and timely problem: stabilizing multi-turn RL for interactive LLM agents.
* The two levels in which T^2PO operates, token and turn level, are intuitive and motivated by practical examples.

Weaknesses:
* The core mechanism is fairly heuristic. In particular, the paper hasn't justified the reasons behind the TTI rule or TDS rule except for the intuitions.
* It is unclear whether the comparisons with baselines are fair or not, because with the turn-level re-generation, T^2PO can use significantly more compute.
* The paper's main text claims that they have "further investigate the sensitivity of \alpha, tolerence threshold, window size, etc.", but it is unclear where in Appendix D can I find these results.

---

> ### Author Rebuttal · Authors · 2026-03-31
>
> > Q1: No reasons for rules.
> >
>
> A1: Thanks. Our intention is not to present TTI and TDS as arbitrary heuristics, but as **two rules of the same principle**: when the uncertainty dynamics stop changing meaningfully, continued exploration is likely to be low-information and mainly inject rollout noise. This motivation is stated at the beginning of Sec 4.2 and 4.3, and is grounded in both the uncertainty analysis in 4.1 and the empirical motivation study in Fig. 4.
>
> **R1.** TTI and TDS are designed tightly around the core **principle**: they do not monitor uncertainty magnitude itself, but rather whether further exploration still changes the model’s internal state in a meaningful way.
>
> **R2.** A single-token uncertainty value is too local and noisy to judge stabilization. Hence TTI uses a trailing-window trend, and TDS uses turn-level aggregation before comparing adjacent turns.
>
> **R3.** The underlying uncertainty signal is not chosen ad hoc: Section 4.1 explains that entropy and confidence have complementary blind spots, so their fused signal is used as a scalar proxy for local distributional stability.
>
> **R4.** The idea is meant to be generalizable because it is an intrinsic control rule over exploration dynamics, not a task-specific action heuristic. We validate it precisely in multi-turn agentic RL, where redundant exploration appears both within a turn and across turns.
>
>
> > Q2: Whether the comparisons are fair?
>
> A2: Thanks. Simply increasing computation does not necessarily improve performance, and without proper control it may even cause training to collapse. In Table 1, *GiGPO + DAPO* applies a computational heavier strategy: whenever the final reward fails to satisfy the diversity criterion, the entire trajectory is oversampled. By contrast, TDS is more lightweight. Despite this larger compute budget, the experimental results show that we consistently outperform the baseline. This strongly suggests that controlling exploration effectively is more helpful. Moreover, Figure 1 shows that uncontrolled additional computation can directly lead to training collapse and thus waste compute. By contrast, both *GiGPO + DAPO* and ours demonstrate that a moderate amount of extra computation is worthwhile when it serves to stabilize training, especially compared with the zero-return regime after collapse.
>
> > Q3: Unclear sensitivity.
>
> A3: Thanks. Appendix D.3 and Table 7 present the analysis on $\alpha$. More study is as follows:
>
> - **Sensitivity analysis on tolerance threshold**.  We observe a consistent pattern: overly small $\varepsilon$ and $\eta$ make the stopping rule too conservative and closer to no intervention, while overly large $\varepsilon$ and $\eta$ make the model more aggressive and can truncate useful reasoning prematurely. The best performance is achieved at a moderate value (10$^{-4}$ for $\varepsilon$ and 10$^{-3}$ for $\eta$), and all tested settings outperform the variant without TTI/TDS, indicating that the method is not overly sensitive to this threshold and admits a reasonably broad working region.
>
> |$\varepsilon$|1e-5|1e-4|5e-4|1e-3|
> |---|---|---|---|---|
> |Score|89.29|93.84|92.96|92.08|
> |Success|78.83|81.64|80.49|80.66|
>
> |$\eta$|1e-4|5e-4|1e-3|1e-2|
> |---|---|---|---|---|
> |Score|88.15|92.87|93.84|91.69|
> |Success|76.27|80.52|81.64|79.36|
>
> - **Sensitivity analysis on window size**. Please refer to A4 for Reviewer E63u.
>
> > Q4: Fragile rule.
>
> A4: Thanks. We do not find this to be a substantive failure mode of TDS. In fact, we observe that many regenerated trajectories correspond to more purposeful action execution. For example, in WebShop, some regenerated trajectories shift from hesitant actions such as repeatedly clicking descriptions to directly taking decisive actions like ``click buy now.”
>
> > Q5: Comparison with DeepConf.
>
> A5: Thanks. We have already discussed the distinction from this line of work in the related work, and we are happy to clarify it more explicitly. The key difference is that we use uncertainty signals to explicitly control the reasoning trajectory online in a fine-grained way. In contrast, DeepConf uses uncertainty primarily as a post hoc filtering signal to discard trajectories judged to be unimportant. Such filtering-based methods typically require oversampling many candidate trajectories before selecting a subset, which is substantially less efficient. We also adapted DeepConf to our setting by incorporating an oversampling-and-filtering mechanism for a direct comparison. Its success rate on WebShop is 72.12, which is substantially lower than the 74.02 achieved by GiGPO + DAPO, and far below our 81.64.
>
> > Q6: Computation cost details.
>
> A6: Thanks. Table 5 already provides the detailed training configuration. Using 8 H100 GPUs, the average training time is 18.3 hours on WebShop, with an average of 6 turns; 27.5 hours on ALFWorld, with an average of 18 turns; and 23.7 hours on SearchQA, with an average of 2 turns. We will also release the full Weights & Biases logs.

---

> > ### Author Rebuttal · Reviewer_A1UR · 2026-04-01
> >
> > Thanks to the authors for their response. It has fully addressed my concerns. It would be helpful if the revised paper could also include the reasoning behind these rules, the sensitivity analysis, and the computation cost. I have increased my score to 4.

---

> > > ### Author Response · Authors · 2026-04-01
> > >
> > > Thank you very much for your recognition and suggestions! We will make the corresponding revisions to the main text.

---

### Decision · Program_Chairs · 2026-04-30

**Decision:**

Accept (spotlight)

**Comment:**

The reviewers agreed that the paper tackles a relevant problem and presents strong, consistent empirical results across multiple benchmarks, including WebShop, ALFWorld, and Search QA. Initially, the reviews raised concerns regarding the heuristic nature of the thresholds, potential entropy collapse, and whether performance gains simply stemmed from increased compute budgets.

The authors provided a strong rebuttal with an ablation study that isolated the method's contribution, alongside entropy curves confirming that the exploration is useful.

While the reliance on threshold-based rules limits the work's algorithmic novelty, it offers a practical and empirically sound solution to a well-known training bottleneck. I recommend a strong accept.